# DLoFT: Gradient-Decoupled Fine-Tuning for Generalizable Long Chain-of-Thought Reasoning

**Sitong Wu**[1,*]  **Haoru Tan**[2,*]  **Jingyao Li**[1]  **Shaofeng Zhang**[3]
**Xiaojuan Qi**[2,†]  **Bei Yu**[1†]  **Jiaya Jia**[1,4,†]

[1]The Chinese University of Hong Kong  [2]The University of Hong Kong
[3]Shanghai Jiao Tong University  [4]The Hong Kong University of Science and Technology

## Abstract

Long chain-of-thought (LongCoT) has emerged as a powerful reasoning paradigm for enabling large language models (LLMs) to solve complex tasks through a systematic and thorough thinking phase. Although supervised fine-tuning (SFT) on high-quality LongCoT traces has proven effective to activate LongCoT abilities, we find that models trained in this way tend to overfit problem-specific knowledge and heuristics, leading to degraded out-of-distribution performance. To address this issue, we propose a Decoupled LongCoT Fine-Tuning (DLoFT) algorithm, which enables the model to learn generalizable LongCoT reasoning abilities while preventing overfitting to the reasoning content with problem-specific information. The key idea is to decouple the gradient into two orthogonal components: 1) a paradigm-relevant gradient corresponding to the general LongCoT paradigm and 2) a content-relevant gradient reflecting the problem-specific information, where only the former gradient is used to update model parameters. Specifically, by leveraging the unique two-phase composition (thinking and solution) of the Long-CoT response, our gradient decoupling mechanism isolates the content-relevant gradient via a projection operation and separates the paradigm-relevant gradient through orthogonalization. Our DLoFT ensures the model concentrate on internalizing the LongCoT paradigm rather than memorizing problem-specific knowledge and heuristics. Extensive experiments demonstrate that our DLoFT significantly improves the generalization behavior of LongCoT abilities compared to SFT while maintaining strong in-distribution performance. The code is available at https://github.com/dvlab-research/DLoFT.

## 1   Introduction

Recent large language models (LLMs), for example, OpenAI o1-o3 [1–3], have achieved remarkable breakthroughs in difficult reasoning tasks [4–6]. A key technique driving these advances is the long chain-of-thought (LongCoT) reasoning paradigm, which encourages models to conduct a systematic and thorough thinking process involving reflection on intermediate steps, error correction, feasibility assessment, and exploration of alternative ideas.

Following these breakthroughs, the community has attempted to replicate such LongCoT reasoning abilities, which remains a significant challenge. Recent efforts typically rely on imitating high-quality LongCoT traces via supervised fine-tuning (SFT) [7–9] or conducting costly reinforcement learning (RL) on tasks with deterministic final answer [10]. Notably, existing works are predominantly focused

---

*Equal contribution (stone-wu@link.cuhk.edu.hk, hrtan@eee.hku.hk)
†Corresponding author

39th Conference on Neural Information Processing Systems (NeurIPS 2025).

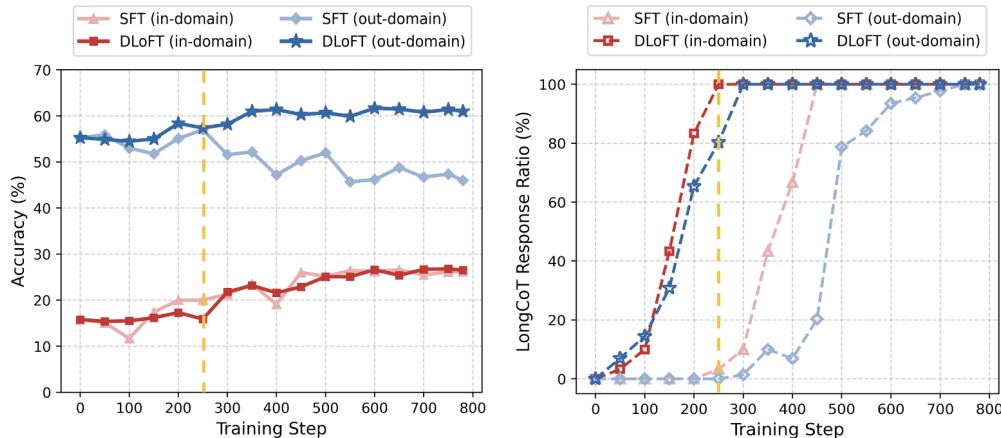

Figure 1: Illustration of the overfitting problem. We train the Qwen2.5-7B-Instruct [11] on OpenR1-Math-5K LongCoT dataset [12] using SFT and our DLoFT respectively, and test on AIME24 math competition [4] as in-domain evaluation and MedQA [13] as out-domain evaluation. (a) shows the change in test accuracy during training, and (b) shows the percentage of the model's response to the test data containing LongCoT behavior as training progresses. The yellow dotted line indicates the turning point where out-domain performance begins to degrade continuously. Please refer to Sec. 3 for more analysis of this overfitting problem.

on narrow fields, such as competition-level mathematics [4], code generation [6], and PhD-level science question answering [5]. Considering the data sparsity and proprietary for some fields, it is non-trivial and impractical to construct a large-scale dataset with comprehensive knowledge coverage, and it is extremely expensive to annotate the high-quality LongCoT response. Consequently, a practical question arises: Can the LongCoT reasoning ability acquired from specific fields (*e.g.*, mathematics) effectively generalize to unseen fields (*e.g.*, engineering, economics, literature, etc)?

**Observations.** To investigate this, we conducted a series of empirical studies as Figure 1. Our experiments reveal a surprisingly poor generalization performance on novel fields. By analyzing the training dynamics, we found that the out-of-distribution performance degradation arises from the overfitting to problem-specific knowledge and heuristics present in the training data. In addition, we have tried common overfitting mitigation strategies, such as early stopping. Although simply applying early stopping produces a slight improvement in unseen fields, it simultaneously leads to insufficient learning in seen fields and does not always exhibit the LongCoT paradigm. We attribute this to the inherent coupling of general LongCoT reasoning abilities with the problem-specific knowledge and heuristics in supervised fine-tuning. These findings underscore the necessity for a training algorithm capable of decoupling problem-specific knowledge from underlying LongCoT reasoning abilities.

**Our Approach.** In this paper, we propose a novel training algorithm, named Decoupled LongCoT Fine-Tuning (DLoFT), which enables the model to master the generalizable LongCoT reasoning paradigm while avoiding overfitting to reasoning content. The central idea is to filter out the gradient component induced from problem-specific information via a gradient decoupling mechanism. Specifically, in each training iteration, we first calculate the full gradient $g_{full}$ of the entire LongCoT response, which blends the influence of two learning signals: one is the LongCoT reasoning paradigm, and the other is the knowledge and skills required to solve the problem. Then, we decouple the full gradient $g_{full}$ into two orthogonal components: (1) the paradigm-relevant gradient $g_{par}$ exclusively capturing the general LongCoT reasoning paradigm, (2) the content-relevant gradient $g_{con}$ corresponding to the problem-specific knowledge and heuristics. A key insight of our decoupling mechanism is that the inherent two-phase structure of LongCoT responses (a thinking phase followed by a solution phase) naturally facilitates the disentanglement of these two gradient components. The thinking phase captures both LongCoT reasoning paradigm and problem-specific information, whereas the solution phase only encodes the latter (see Sec. 4.1 and 4.2 for more explanation). Therefore, by comparing the gradients induced by the full LongCoT response and the solution phase alone, we can isolate the gradient $g_{con}$ using a projection operation and separate $g_{par}$ via an orthogonalization operation (elaborated in Sec. 4.3). Finally, the model is updated using only the paradigm-relevant gradient com-

ponent. This ensures the model focuses on internalizing the LongCoT paradigm without memorizing problem-specific knowledge and heuristics. The overall algorithm is outlined in Algorithm 1.

We compare our DLoFT with SFT on the challenging benchmarks in 17 diverse domains for both generalist and domain-specific LLMs. Extensive experiments demonstrate that: (1) our DLoFT effectively mitigates the overfitting problem of SFT and learns stronger generalizable LongCoT reasoning ability. For example, it significantly outperforms SFT by an average of +16.4 points on out-domains and has a notable advantage of +11.5 points on in-domains. (2) As a cold-start stage for RL, our DLoFT can free the cold-start from the dependence on in-domain LongCoT data, and serves as a stronger foundation for subsequent RL, amplifying the outcome of RL.

## 2 Related Work

### 2.1 LLM Reasoning from CoT to LongCoT

Reasoning is a core capability of large language models (LLMs), reflecting their ability to address complex real-world problems. Since 2022, Chain-of-Thought (CoT) [14] has enhanced the ability of LLM on reasoning tasks (like mathematics [15] and basic programming [16]) by encouraging them to generate intermediate steps before deriving a final answer. While LLM with CoT reasoning paradigm still struggles with highly complex reasoning tasks like competition-level math [4], coding competitions [6], and PhD-level science QA [5]. For example, GPT-4o [17] only achieves 13.4% accuracy on the well-known math competition AIME24 [4]. The release of OpenAI's o1 model [1] broke this status and made great progress in these tasks. The underlying reason for o1's success is the shift in the reasoning paradigm from a short-formed CoT to a more comprehensive Long Chain-of-Thought (LongCoT), which scales up the inference computation of LLMs by engaging in a deeper thinking phase before delivering the final answer. Unlike a deterministic solution, this thinking process allows the model to freely and thoroughly explore a given problem, including several typical characteristics: (1) exploring different ideas, (2) backtracking to prior steps if the current step is hopeless, (3) reflecting the correctness of previous steps, and (4) correcting errors.

### 2.2 Techniques for Enabling LongCoT Reasoning

Following the breakthrough of o1 [1], the community has attempted to cultivate the LongCoT reasoning abilities for LLMs. These efforts can be categorized into four groups:

**Inducing Reflection via Inference-Time Intervention.** The first line of work encourages the model to engage in self-reflection through an inference-time intervention mechanism. For example, [7] demonstrated that simply inserting the token "Wait" into the model's current reasoning trace can effectively prompt the model to reflect on its current step. [18] further revealed that this self-reflection behavior is already acquired during the pre-training phase, which offers a plausible explanation for the effectiveness of inference-time intervention techniques. However, inference-time intervention is less practical because it relies on manual intervention to trigger reflective behavior rather than spontaneously performing self-reflection and correction when necessary (like o1 [1]).

**Imitate High-Quality LongCoT Responses via Supervised Fine-Tuning (SFT).** Inspired by knowledge distillation [19, 20], the most straightforward way to enable LongCoT abilities is fine-tuning the LLM to imitate the high-quality LongCoT reasoning traces, which can be curated in two ways: one is to collect the responses from existing models with strong LongCoT abilities (e.g., DeepSeek-R1 [10] and QwQ [21, 22], etc.), the other is to manually synthesize the tree search sampling trajectories of CoT model into the LongCoT form [23, 24]. Existing studies narrowly focus on a few topics for both training and evaluation, including competition-level mathematics [4], code generation [6], and PhD-level science question answering [5]. In contrast, our experiments reveal that SFT on LongCoT data with narrow topics leads to poor generalization on diverse novel topics, which is caused by the fact that the model is prone to overfitting the problem-specific knowledge and skills involved in the training data. To address this issue, we propose a Decoupled Fine-Tuning (DLoFT) algorithm for learning generalizable LongCoT reasoning, which decouples the gradient exclusively corresponding to general LongCoT reasoning paradigm and the gradient caused by problem-specific information.

**LongCoT Emerges through Zero Reinforcement Learning (Zero-RL).** DeepSeek-R1-Zero [10] first found that LongCoT reasoning abilities can naturally emerge through reinforcement learning (RL) with rule-based rewards for a base LLM, referred to as Zero-RL training. In particular, they experimented on a 671B model [25] and demonstrated that the model naturally learned to solve reasoning tasks with more thinking time (increased average response length) and to reflect using an anthropomorphic tone (called "aha moment"). Subsequent works [26–28] have attempted to reproduce its performance and emergence phenomenon by demystifying its proprietary training data and settings.

**Reinforcement Learning (RL) with LongCoT Cold-start.** Despite the success of DeepSeek-R1-Zero[10], recent studies [29] have found that Zero-RL does not always stably emerge LongCoT reasoning paradigm, particularly in terms of extended reasoning length and complexity. A SFT cold-start phase on LongCoT-style data is essential to first instill the LongCoT reasoning paradigm in the model before applying RL for further enhancement. This cold-start phase is proven to make further RL improvement easier [10, 29]. Under this "RL with Cold-start" training scheme, our DLoFT algorithm can effectively replace the SFT in the cold-start phase. Experiments show that RL with our DLoFT cold-start achieves better generalization and superior RL gains compared to SFT counterpart.

## 3 Overfitting Problem Investigation

From Figure 1, we can observe that supervised fine-tuning (SFT) on LongCoT data involving a few specific domains leads to notable performance drops on 11 out-domain benchmarks, with degradations ranging from -1.9 to -16.0 points. This clearly highlights the poor generalization behavior of standard LongCoT Supervised Fine-Tuning (SFT).

To further investigate the underlying reason for this out-of-distribution generalization problem, we monitor two key metrics throughout the training process on both in-domain and out-domain benchmark: (1) the accuracy; (2) the percentage of model's response to the test data that exhibits LongCoT behavior. In particular, we use GPT-4o-mini API to detect whether the model response contains LongCoT behaviors such as reflection, error correction, and exploration of new ideas.

As shown in Figure 1, although the accuracy on the in-domain benchmark steadily improves during training, the accuracy on the out-domain benchmark initially increases slightly (from 55.3 to 57.0) and then begins to consistently decline after 250 step, eventually dropping to 46.9. This trend suggests that the out-of-distribution performance degradation is due to the model overfitting to problem-specific knowledge and heuristics present in the training data.

In addition, common overfitting mitigation strategy, such as early stopping (stopping training before out-domain performance degradation), has been shown to be ineffective. Although simply applying early stopping can prevent a performance degradation on out-domain benchmark, it leads to both the limited performance gains and insufficient learning on the LongCoT reasoning paradigm. As shown in Figure 1(a), if training is stopped early at step 250 (indicated by the yellow dashed line), we forfeit approximately 7 points on the accuracy of in-domain benchmark. Simultaneously, as Figure 1(b), only 3.3% of the in-domain test examples exhibit LongCoT behavior at the early stopping moment, indicating that the model has not yet sufficiently internalized the LongCoT reasoning paradigm.

These observations underscore a key challenge: naive supervised fine-tuning (SFT) is prone to overfit to problem-specific knowledge and heuristics present in the training data, rather than internalizing a generalizable LongCoT reasoning paradigm. In contrast, our DLoFT not only ensures a steady increase in in-domain and out-domain performance (as Figure 1(a)), but also demonstrates faster learning of the LongCoT paradigm (see Figure 1(b)).

## 4 Method

In this section, we introduce our Decoupled LongCoT Fine-Tuning (DLoFT) algorithm, which enhances the generalizability of learned LongCoT reasoning abilities while preventing overfitting to specific problems in training data. First, we review the characteristics of LongCoT data in Sec. 4.1, providing the necessary background for understanding the motivation and algorithm. Then, in Sec. 4.2, we discuss the challenges in LongCoT fine-tuning, and highlight our motivation for leveraging the unique structure of LongCoT data. Finally, we elaborate our algorithm in Sec. 4.3.

## 4.1 Preliminary: LongCoT Reasoning Format

The LongCoT-style response is structured into two phases: (1) **Phase-1**: a systematic *thinking* phase that includes analyzing problems, summarizing findings, exploring new ideas, reflecting on correctness, and correcting errors. (2) **Phase-2**: a *solution* phase that presents a deterministic step-by-step solution derived from the successful attempts during the thinking phase. This two-phase structure in LongCoT enables a more thorough, exploratory and reflective reasoning process, in contrast to the conventional CoT which typically follows a single-phase structure focused solely on step-by-step reasoning. Figure 5 in Appendix provides an example of LLM's response under LongCoT Reasoning.

## 4.2 Motivation

LongCoT fine-tuning aims to enable the model to adopt LongCoT reasoning paradigm on problem solving: first engaging in deliberate thinking to make a deep and structured exploration of the problem, and then synthesizing a deterministic solution based on the successful reasoning traces.

The mainstream training objective is to directly imitate high-quality LongCoT responses on thousands of representative reasoning problems [7, 8]. This optimization objective inherently couples two types of learning signals: (1) **reasoning paradigm**, *i.e.*, how to structure thoughts, analyze problem, reflect on previous steps, correct errors, assess feasibility, backtrack to prior step, and brainstorm alternative ideas; (2) **reasoning content**, *i.e.*, the relevant knowledge and skills required for solving the problem. Typically, models require more training iterations to internalize strong and stable LongCoT abilities due to the complexity of the LongCoT reasoning process. However, this increases the risk of overfitting — not to the reasoning paradigm itself, but to the problem-specific knowledge and skills covered in the training data. As a result, the model learns to reason effectively within seen topics, but struggles to generalize to unseen ones.

To address this issue, we propose encouraging the model to focus on learning the LongCoT reasoning paradigm, rather than memorizing problem-specific knowledge or heuristics. Concretely, we decouple the gradients into two orthogonal components: one corresponding to problem-specific content and the other capturing exclusive LongCoT reasoning paradigm. Only the latter gradient component is adopted to update model parameters.

A key insight of our work is that the unique two-phase structure of LongCoT responses naturally facilitates the decoupling of paradigm-relevant and content-relevant gradients. As described in Sec. 4.1, the thinking phase encodes both the reasoning paradigm and problem-specific content, while the solution phase reflects only the latter. By comparing the gradients induced by the full LongCoT response and the solution part alone, we can isolate the gradient component that exclusively corresponds to the LongCoT reasoning paradigm. The algorithm details are elaborated in the following sub-section.

## 4.3 Decoupled LongCoT Fine-Tuning

This section provides a detailed introduction to our training methodology designed to enhance the long reasoning abilities of large language models, which we name Gradient-Decoupled Fine-Tuning. Our core idea is a gradient decoupling mechanism that explicitly separates the supervisory signals related to long reasoning from those that are purely problem-specific. This approach fundamentally differs from conventional Supervised Fine-Tuning (SFT), where the model's learning objective is typically to replicate or memorize the provided annotated long reasoning process and the final answer. The overall procedure is outlined in Algorithm 1. The core of the entire optimization algorithm consists of the following three sequential steps, used for gradient decoupling and parameter updating.

**Step 1: Compute the Full Response Gradient**

For each training example in the mini-batch, represented as a tuple $(P_i, T_i, S_i)$, where $P_i$ denotes the problem, $T_i$ signifies the exploratory thinking process, and $S_i$ indicates the deterministic solution, we input the question $P_i$ into the model $\boldsymbol{\theta}$. We then compute the negative log-likelihood (NLL) loss over the complete Long Chain-of-Thought (LongCoT) response, represented as $T_i \oplus S_i$, where $\oplus$ denotes the concatenation of the two text components. Formally, the gradient $\mathbf{g}_{\text{full}}$ of the averaged

**Algorithm 1:** Decoupled LongCoT Fine-Tuning (DLoFT)

---

**Input:** Training dataset $\mathcal{D}$, initial model parameters $\boldsymbol{\theta}$, learning rate $\eta$, maximum iterations $\mathcal{T}$, batch size $\mathcal{B}$

**for** $t = 1$ **to** $\mathcal{T}$ **do**

    Sample a mini-batch $\{(P_i, T_i, S_i)\}_{i=1}^{\mathcal{B}}$ from $\mathcal{D}$, where

    $\begin{cases} P_i : \text{the } i\text{-th problem} \\ T_i : \text{the exploratory thinking process in LongCoT response for } P_i \\ S_i : \text{the deterministic solution in LongCoT response for } P_i \end{cases}$ ;

    **Step 1:** Compute the Full Response $(T_i \oplus S_i)$ Gradient        // $\oplus$: concatenation

      $\mathcal{L}_{\text{full}} = -\frac{1}{\mathcal{B}} \sum_{i=1}^{\mathcal{B}} \sum_{k=1}^{|T_i \oplus S_i|} \log p_{\boldsymbol{\theta}} \big( (T_i \oplus S_i)^k \mid (T_i \oplus S_i)^{<k}, P_i \big)$;     // NLL loss

      $\mathbf{g}_{\text{full}} = \nabla_{\boldsymbol{\theta}} \mathcal{L}_{\text{full}}$;                         // compute gradient

    **Step 2:** Gradient Decoupling

     ▷ Compute the reference gradient (*w.r.t.* problem-specific information):

       $\mathcal{L}_{\text{ref}} = -\frac{1}{\mathcal{B}} \sum_{i=1}^{\mathcal{B}} \sum_{k=1}^{|S_i|} \log p_{\boldsymbol{\theta}} \big( S_i^k \mid S_i^{<k}, P_i \big)$;     // NLL loss with $S_i$ as target

       $\mathbf{g}_{\text{ref}} = \nabla_{\boldsymbol{\theta}} \mathcal{L}_{\text{ref}}$;                        // compute gradient

     ▷ Decouple the content-relevant gradient (*w.r.t.* problem-specific information) from $\mathbf{g}_{\text{full}}$:

       $\mathbf{g}_{\text{con}} = \frac{\langle \mathbf{g}_{\text{full}}, \mathbf{g}_{\text{ref}} \rangle}{\|\mathbf{g}_{\text{ref}}\|^2} \cdot \mathbf{g}_{\text{ref}}$;             // projection

     ▷ Decouple the paradigm-relevant gradient (*w.r.t.* LongCoT reasoning paradigm) from $\mathbf{g}_{\text{full}}$:

       $\mathbf{g}_{\text{par}} = \mathbf{g}_{\text{full}} - \mathbf{g}_{\text{con}}$;                  // orthogonalization

    **Step 3:** Update Model Parameters

    $\boldsymbol{\theta} \leftarrow \boldsymbol{\theta} - \eta \cdot \mathbf{g}_{\text{par}}$;           // use decoupled paradigm-relevant gradient to update model

**return** $\boldsymbol{\theta}$;

---

loss function $\mathcal{L}_{\text{full}}$ for this mini-batch is defined as:

$$\mathbf{g}_{\text{full}} = \nabla_{\boldsymbol{\theta}} \mathcal{L}_{\text{full}}, \quad \text{where} \quad \mathcal{L}_{\text{full}} = -\frac{1}{\mathcal{B}} \sum_{i=1}^{\mathcal{B}} \sum_{k=1}^{|T_i \oplus S_i|} \log p_{\boldsymbol{\theta}} \Big( (T_i \oplus S_i)^k \Big| (T_i \oplus S_i)^{<k}, P_i \Big), \quad (1)$$

where the gradient contains two intertwined types of supervisory information. First, there is problem-specific content, which includes the facts, formulas, or procedures needed for a particular problem. Focusing too much on this during training can cause the model to memorize details, limiting its ability to develop generalizable reasoning skills. Second, there is information aimed at activating the model's LongCoT abilities. This involves guiding the model to generate and follow logical steps to reach a solution, rather than just memorizing input-output pairs. This approach is crucial for enabling the model to generalize beyond specific training examples and apply learned reasoning patterns to new and unseen problems.

### Step 2: Gradient Decoupling via Projection and Orthogonalization

In this step, we decouple the full response gradient $\mathbf{g}_{\text{full}}$ defined by Eq.(1) into the *paradigm-relevant gradient* and the *content-relevant gradient*. The former contains supervisory information for long reasoning abilities that is beneficial for generalization, while the latter contains problem-specific supervisory information.

**Compute the reference gradient.** To isolate the content-specific gradient from $\mathbf{g}_{\text{full}}$, we first compute a reference gradient that captures only the problem-specific content. For each training example $(P_i, T_i, S_i)$, we evaluate the negative log-likelihood (NLL) loss solely over the solution component $S_i$, defined as follows:

$$\mathbf{g}_{\text{ref}} = \nabla_{\boldsymbol{\theta}} \mathcal{L}_{\text{ref}}, \quad \text{where} \quad \mathcal{L}_{\text{ref}} = -\frac{1}{\mathcal{B}} \sum_{i=1}^{\mathcal{B}} \sum_{k=1}^{|S_i|} \log p_{\boldsymbol{\theta}} \Big( S_i^k \Big| S_i^{<k}, P_i \Big). \quad (2)$$

This gradient, $\mathbf{g}_{\text{ref}}$, signifies how the model would adjust its parameters if it were trained to directly produce the final deterministic solution $S_i$, without undergoing an extensive exploratory reasoning phase before generating the answer. Consequently, $\mathbf{g}_{\text{ref}}$ serves as a reference gradient for problem-specific reasoning content, as the NLL loss target $S_i$ encompasses only the knowledge and heuristics pertinent to the problem, devoid of any LongCoT reasoning framework.

**Decouple the content-relevant gradient via projection.** Using the reference gradient $\mathbf{g}_{\text{ref}}$, which focuses on problem-specific reasoning, we can extract the content-relevant component from the full gradient $\mathbf{g}_{\text{full}}$ through a straightforward projection operation. The projection is defined as follows:

$$\mathbf{g}_{\text{con}} = \frac{\langle \mathbf{g}_{\text{full}}, \mathbf{g}_{\text{ref}} \rangle}{\|\mathbf{g}_{\text{ref}}\|^2} \cdot \mathbf{g}_{\text{ref}}, \tag{3}$$

where $\langle \cdot, \cdot \rangle$ denotes the inner product and $\| \cdot \|^2$ represents the $\ell_2$ norm. Conceptually, $\mathbf{g}_{\text{ref}}$ defines the subspace that captures the influence of problem-specific content on model updates. By projecting $\mathbf{g}_{\text{full}}$ onto $\mathbf{g}_{\text{ref}}$, we isolate the component of $\mathbf{g}_{\text{full}}$ that pertains exclusively to problem-specific reasoning. This results in $\mathbf{g}_{\text{con}}$, which represents the content-relevant gradient component.

**Decouple the paradigm-relevant gradient via orthogonalization.**
After isolating the content-relevant gradient $\mathbf{g}_{\text{con}}$, we extract the paradigm-relevant component by removing the influence of problem-specific reasoning from the full gradient $\mathbf{g}_{\text{full}}$. This is accomplished through an orthogonalization operation defined as:

$$\mathbf{g}_{\text{par}} = \mathbf{g}_{\text{full}} - \mathbf{g}_{\text{con}}. \tag{4}$$

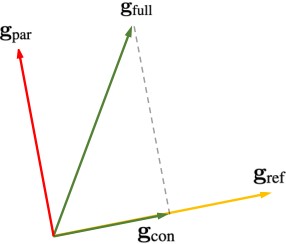

Figure 2: Relationships between the involved gradients in DLoFT.

By subtracting $\mathbf{g}_{\text{con}}$ from $\mathbf{g}_{\text{full}}$, we eliminate the component associated with problem-specific reasoning, resulting in a purified gradient $\mathbf{g}_{\text{par}}$ that is orthogonal to $\mathbf{g}_{\text{con}}$. This ensures that $\mathbf{g}_{\text{par}}$ captures aspects of $\mathbf{g}_{\text{full}}$ that are independent of problem-specific knowledge. Consequently, $\mathbf{g}_{\text{par}}$ represents the paradigm-relevant gradient, focused solely on the LongCoT reasoning paradigm. It indicates how the model should adjust its parameters to enhance its LongCoT reasoning abilities, irrespective of the specific problems present in the training data. Figure 2 illustrates the relationships between these gradients.

**Step 3: Update the Model with Paradigm-Relevant Gradient**

To avoid the model overfitting to the problem-specific knowledge involved in training data, we drop the content-relevant gradient $\mathbf{g}_{\text{con}}$ and update the model using only the paradigm-relevant gradient $\mathbf{g}_{\text{par}}$:

$$\boldsymbol{\theta} \leftarrow \boldsymbol{\theta} - \eta \cdot \mathbf{g}_{\text{par}}, \tag{5}$$

where $\eta$ is the learning rate. This gradient decoupling mechanism ensures that the model concentrately internalizes generalizable LongCoT reasoning abilities rather than memorizing problem-specific information in training data.

## 5   Experiments

In this section, we first introduce the experimental setups in Sec. 5.1. Then, in Sec. 5.2, we demonstrate the superiority of our DLoFT algorithm over supervised fine-tuning (SFT), both as a standalone supervised learning stage and as a cold-start stage prior to reinforcement learning (RL). Ablation studies on our key designs and further discussions are detailed in Appendix A.

### 5.1   Experimental Setup

**Models.** To evaluate the effectiveness and generality of our DLoFT algorithm, we conduct experiments on both generalist and domain-specific LLMs with varying sizes. For generalist LLMs, we adopt the Qwen2.5-Instruct series [11] at 7B and 32B scales due to their strong performance across a variety of tasks. For domain-specific LLMs, we consider three major domains: math, code, and medicine. Concretely, we use Qwen2.5-Math-7B-Instruct [30] for mathematical reasoning, Qwen2.5-Coder-7B-Instruct [31] for code generation, and Meditron-7B [32] for medical reasoning.

**Evaluation.** To investigate the comprehensive generalization, we conduct evaluation on a variety of benchmarks across 20 domains. For major domains with many well-known benchmarks, we test on the most commonly-used and challenging ones, such as the competition-level AIME24 [4], MATH-500 [33] and OlympiadBench [34] for mathematics, LiveCodeBench (v2) [6] for code generation, and MedQA [13] for medical reasoning. For other less-studied domains with few

well-established benchmarks, we directly use the corresponding subset from SuperGPQA [35], a benchmark focused on graduate-level reasoning problems, as our test set. These domains cover STEM disciplines, social sciences, and humanities, including physics, chemistry, biology, computer, mechanical, electronic, communication, astronomy, geography, civil, agriculture, economics, history, law, literature, philosophy, and sociology.

**Training Datasets.** Our experiments utilize two types of LongCoT datasets that differ in coverage:

- *Mixed-Domains Dataset*. We adopt the well-known s1K [7] dataset due to its good trade-off between effectiveness and efficiency. It contains one thousand high-quality data covering mathematics, programming, science, and puzzles, and is carefully curated based on quality, difficulty, and diversity simultaneously. Each data has a LongCoT response generated by Google Gemini Flash Thinking model [36].

- *Single-Domain Dataset*. We use open-source datasets that focus on three major domains: math, code, and medicine. Specifically, we take the OpenR1-Math-220K dataset [12] for mathematics, the programming-related subset from OpenThoughts-114K dataset [37] for code, and the Medical-o1 dataset [38] for medicine. Prior studies [39, 7] have revealed that learning LongCoT reasoning capabilities does not require hundreds of thousands of data, because it focuses on changing the reasoning paradigm rather than memorizing a large amount of knowledge. Therefore, we randomly sample 5K data from each of the above three datasets for our training, denoted as OpenR1-Math-5K, OpenThoughts-Code-5K, and Medical-o1-5K, respectively.

**Training Settings.** For fair comparison, we use the same training settings for SFT and our DLoFT. The models are trained for 10 epochs using the AdamW optimizer with weight decay of zero. The learning rate is first increased from 0 to 1e-5 with a warm-up ratio of 0.03, and then decreased following a cosine decay schedule. We set the batch size as 16 when training on s1K [7] dataset, and 64 for other training datasets. During the RL training stage, we use the recent popular GRPO algorithm [40] with a KL penalty coefficient of 0.001. The batch size is set to 128, and the learning rate is set to 1e-6 keeping constant throughout the training. We train the model for 3 epochs, because it is found that the reward curve converges after about 3 epochs.

## 5.2 Main Results

### 5.2.1 DLoFT Generalizes Better than SFT in all Evaluation Settings

**Results on Generalist LLMs.** Figure 3 compares our DLoFT with SFT on two generalist LLMs (Qwen2.5-7B/32B-Instruct). We evaluate across 17 diverse domains and report the relative accuracy compared to the baseline model to intuitively illustrate the gain and degradation in performance after training. It can be observed that SFT suffers from clear overfitting to training topics, leading to notable performance drops on out-of-distribution domains. For instance, SFT causes a degradation of -16.0 points on "communication" for 7B model. In stark contrast, our DLoFT consistently improves performance across both in-domain and out-of-domain benchmarks, demonstrating a clear advantage in generalization ability. For example, compared with SFT, our DLoFT exhibits an advantage of +12.4 and 16.4 points on 7B and 32B models. This shows that our method not only mitigates overfitting but also promotes the transferability of LongCoT reasoning ability across various domains. Notably, DLoFT still outperforms SFT even on in-domain benchmarks, yielding additional +8.9 and +11.5 accuracy gains for 7B and 32B models, respectively. This reveals that the gain brought by SFT is still limited due to overfitting to narrow problem-specific content in the training set with narrow topics, which hinders its capacity to generalize even within the same domain. Averaged over all 17 domains, our DLoFT achieves a substantial improvement of +9.8 for Qwen2.5-7B-Instruct, while SFT leads to a degradation of -1.8. In summary, these results indicate that our DLoFT enables generalist LLMs to efficiently acquire generalizable LongCoT capabilities using only a small amount of representative data, without the necessity to construct a large-scale dataset that exhaustively covers a wide range of topics. This not only significantly reduces the cost of data collection, curation, and annotation, but also greatly saves the required training time and computational resources.

**Results on Domain-specific LLMs.** DLoFT's strong generalization can also be observed in the case of domain-specific LLMs. Figure 4 compares our DLoFT and SFT on three excellent domain-specific LLMs trained with either in-domain or out-domain LongCoT data. Results show that our DLoFT consistently outperforms SFT across all evaluation settings, regardless of whether the training data is in-domain or out-domain. In particular, when models are trained on out-domain LongCoT data,

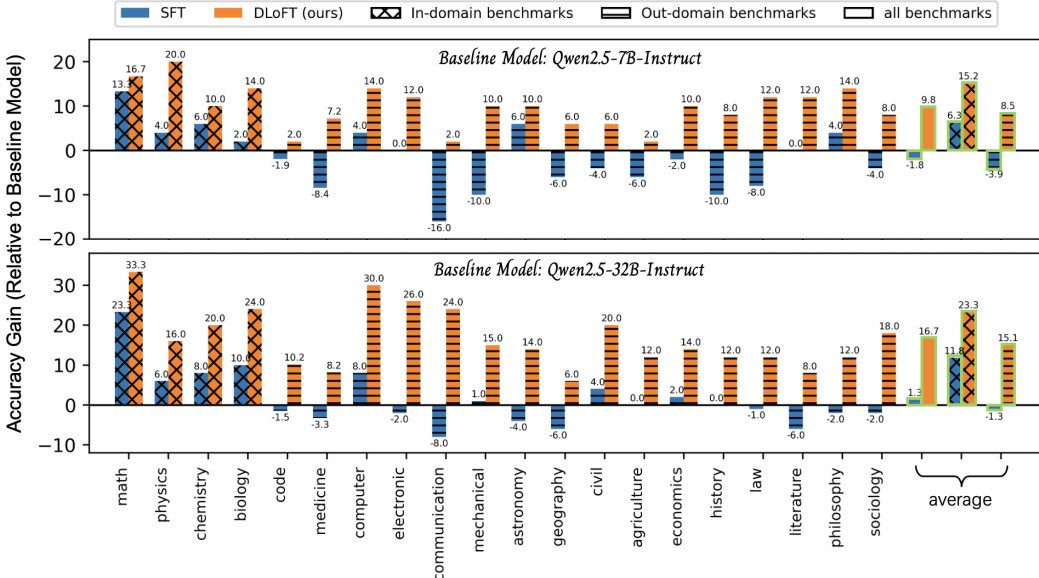

Figure 3: Comparison between SFT and our DLoFT on two generalist LLMs trained on s1K dataset [7] and evaluated on 17 domains. The last three columns with green edges show the average gain on these domains for all benchmarks, in-domain benchmarks, and out-domain ones, respectively. To intuitively observe which domains experienced performance degradation after training, we only report the relative performance (pass@1 accuracy) compared to the baseline model. It can be found that SFT leads to significant performance degradation on out-of-distribution domains, while our DLoFT boosts the performance of baseline model on both in-domain and out-domain benchmarks.

SFT hurts the performance by -3.3, -1.6, and -7.5 points for these three baseline models in math, code, and medicine domains, respectively. In contrast, our DLoFT achieves substantial improvements accordingly: +10.0 on AIME24 (math), +1.8 on LiveCodeBench (code), and +9.1 on MedQA (medical). Interestingly, our DLoFT shows comparable performance when training on in-domain or out-domain data, which indicates that our DLoFT algorithm effectively isolates the influence related to LongCoT reasoning paradigm from the training data, while being robust to domain-specific information that may otherwise act as confounding noise. Moreover, even when in-domain LongCoT data is available, our DLoFT still provides moderate improvements over SFT. We attribute this to SFT's tendency to focus on domain- and problem-specific knowledge or skills present in the training data, thereby limiting its capacity to learn general reasoning ability. In contrast, by decoupling domain-specific signals from reasoning ability signals, our DLoFT concentrates on learning general reasoning ability, leading to more robust gains on in-domain benchmarks as well. In summary, these results demonstrate that our DLoFT offers an efficient and practical way for enabling LongCoT reasoning in domain-specific LLMs. With DLoFT, a domain LLM can activate its LongCoT capability by leveraging publicly available LongCoT data from other domains, freed from constructing its own in-domain LongCoT datasets costly. This not only reduces the expense of data annotation and training, but also provides a fundamentally viable solution for domains with privacy constraints (data cannot be annotated through public APIs) or domains whose data is intrinsically difficult to collect.

#### 5.2.2 DLoFT is a More Effective LongCoT Cold-Start for Reinforcement Learning (RL)

We highlight two key benefits when using our DLoFT as a cold-start training algorithm before RL:

**Relax the Dependence on In-domain Data in Cold-Start.** It can be observed from Figure 4 that SFT cold-start relies heavily on in-domain LongCoT data. The performance degrades consistently when performing SFT cold-start on out-domain LongCoT data. RL may not always be able to completely recover the performance loss caused by the SFT cold-start, so that the final performance after RL may be worse than the baseline. For example, as Figure 4c, SFT+RL still falls behind the baseline model by 4.9 points. In contrast, DLoFT frees the cold-start from the indispensability for in-domain

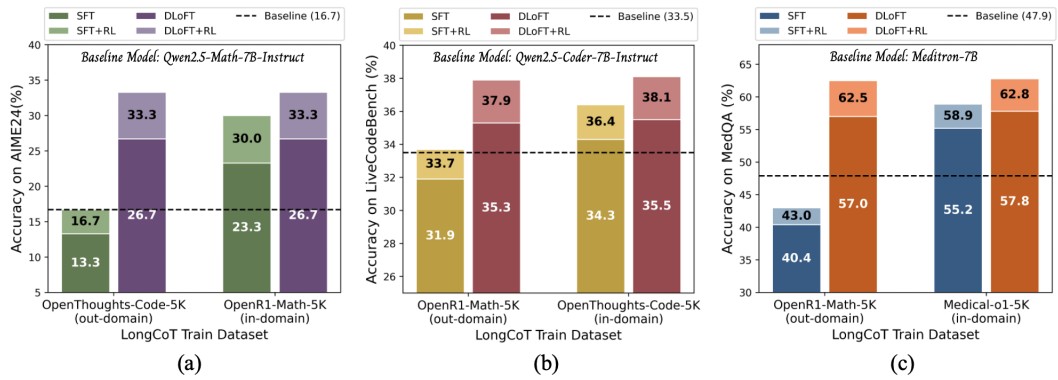

Figure 4: Compare SFT with our DLoFT on three domain-specific LLMs. Models are trained on out-domain and in-domain LongCoT data, respectively. In contrast to the degraded performance after SFT on out-domain LongCoT data, our DLoFT shows much better generalization and further promotes the subsequent reinforcement learning (RL).

LongCoT data. Results show that our DLoFT cold-start, using publicly available LongCoT data from other domains, can achieve performance comparable to that obtained with an equal amount of in-domain data. Such strong generalization and robustness bring great convenience for the data collection and curation in the cold-start stage.

**Amplify the Outcome of RL.** Even with access to in-domain LongCoT cold-start data, our DLoFT offers a stronger foundation for subsequent RL, boosting RL to achieve higher gains. For example, as Figure 4c, RL after SFT cold-start brings a +3.7 gain, while RL after DLoFT cold-start achieves a larger +5.0 improvement. One reasonable explanation is that SFT cold-start, even when performed on in-domain data, may still introduce slight overfitting to domain-specific information, which provides a suboptimal foundation for the subsequent RL stage and increases the burden for further improvement.

## 6 Conclusion

We introduced Decoupled LongCoT Fine-Tuning (DLoFT), an algorithm designed to endow LLMs with generalizable LongCoT reasoning abilities while mitigating overfitting to problem-specific information. By decoupling gradients into paradigm-relevant and content-relevant components using the intrinsic two-phase structure of LongCoT responses, DLoFT updates the model solely with the reasoning-relevant signal. Our approach preserves in-distribution performance and yields substantial gains in out-of-distribution generalization, offering a robust and efficient path toward scalable and efficient LongCoT activation across diverse tasks and domains. We compare our DLoFT with SFT on the challenging benchmarks in 17 diverse domains for both generalist and domain-specific LLMs. Extensive experiments demonstrate that: (1) our DLoFT effectively avoids the overfitting problem in SFT and learns stronger generalizable LongCoT reasoning ability. (2) As a cold-start stage prior to RL, our DLoFT can free the cold-start from the dependence on in-domain LongCoT data, and serves as a stronger foundation for subsequent RL, amplifying the gain of RL.

## Acknowledgements

This work was supported in part by the Research Grants Council under the Areas of Excellence scheme grant AoE/E-601/22-R, Hong Kong Research Grant Council - Early Career Scheme (Grant No. 27209621), General Research Fund Scheme (Grant No. 17202422, 17212923, 17215025), Theme-based Research (Grant No. T45-701/22-R) and Shenzhen Science and Technology Innovation Commission (SGDX20220530111405040). Part of the described research work is conducted in the JC STEM Lab of Robotics for Soft Materials funded by The Hong Kong Jockey Club Charities Trust.

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

# A Ablation and Further Study

## A.1 Effect of Projection Operation in Eq. (3)

As mentioned in Sec. 4.3 in the main paper, our DLoFT first derive a reference gradient $\mathbf{g}_{\text{ref}}$ that captures only the problem-specific information, and then use a projection operation to isolate the component of the full gradient $\mathbf{g}_{\text{full}}$ that aligns with $\mathbf{g}_{\text{ref}}$, resulting in the content-relevant gradient $\mathbf{g}_{\text{con}}$ (as defined in Eq. (3) in the main paper). Here, we ablate the effect of this projection operation by comparing the following two settings: (1) using the projection as Eq. (3), that is, $\mathbf{g}_{\text{con}} = \frac{\langle \mathbf{g}_{\text{full}}, \mathbf{g}_{\text{ref}} \rangle}{\|\mathbf{g}_{\text{ref}}\|^2} \cdot \mathbf{g}_{\text{ref}}$; (2) skipping the projection and directly assigning $\mathbf{g}_{\text{con}} = \mathbf{g}_{\text{ref}}$. As shown in Table 1, omitting the projection still achieves better performance than standard supervised fine-tuning (SFT) and partially mitigates the out-of-domain performance drop. However, the variant without projection consistently underperforms the full DLoFT algorithm with the projection operation. This result indicates that the projection step plays a critical role in more precisely isolating problem-specific gradients, and its removal leads to incomplete decoupling, thereby reducing the effectiveness of the filtering problem-specific knowledge and heuristics.

Table 1: Ablation on the effect of projection operation in Eq. (3) in the main paper. The first line corresponds to the model before training. We train the Qwen2.5-7B-Instruct [11] model on s1K [7] dataset and report the pass@1 accuracy average on four in-domain benchmarks and 16 out-domain benchmarks, respectively.

| Method | Projection | In-domain Acc@1 (avg) | Out-domain Acc@1 (avg) |
|--------|------------|-----------------------|------------------------|
| - | - | 21.5 | 31.2 |
| SFT | - | 27.8 | 27.3 |
| DLoFT | ✗ | 30.3 | 31.8 |
|  | ✔ | 36.7 | 39.7 |

## A.2 Effectiveness of Our Gradient Decoupling Mechanism

The gradient decoupling mechanism in our DLoFT aims to decouple the full gradient into two orthogonal components: (1) the paradigm-relevant gradient $\mathbf{g}_{\text{par}}$ exclusively capturing the general LongCoT reasoning paradigm; (2) the content-relevant gradient $\mathbf{g}_{\text{con}}$ corresponding to the problem-specific knowledge and heuristics. In order to verify the effectiveness of our decoupling mechanism, we conduct a comparative study between updating the model using only the content-relevant gradient $\mathbf{g}_{\text{con}}$ versus only the paradigm-relevant gradient $\mathbf{g}_{\text{par}}$. The underlying intuition is that if the decoupling mechanism is effective, then $\mathbf{g}_{\text{con}}$ should not contain any signal related to the LongCoT paradigm. As a result, updating the model using only $\mathbf{g}_{\text{con}}$ should not lead to any emergence of LongCoT-style behavior. To test this, we train two separate models: one using $\mathbf{g}_{\text{par}}$ and one using $\mathbf{g}_{\text{con}}$, with all other settings held constant. We then measure the percentage of generated responses that exhibit LongCoT behavior on both in-domain and out-of-domain benchmarks. As shown in Table 2, the model updated with $\mathbf{g}_{\text{par}}$ demonstrates clear learning of LongCoT behavior and achieves strong generalization, while the model trained with $\mathbf{g}_{\text{con}}$ fails to produce any LongCoT-style reasoning and performs close to the baseline. This sharp contrast provides strong empirical evidence that our decoupling effectively isolates the paradigm-relevant gradient, confirming both the functional accuracy and practical value of our DLoFT approach.

## A.3 Effect of Weight Decay Regularization

A natural method for preventing overfitting is the use of weight decay, a widely adopted regularization technique in model training. However, we find that weight decay is largely ineffective in preventing the type of overfitting observed in LongCoT supervised fine-tuning. Specifically, as shown in Table 3, although applying weight decay can reduce the model's tendency to memorize problem-specific details, it does not improve generalization to out-of-distribution problems. This is because weight decay regularizes model parameters globally without distinguishing between gradients that encode

Table 2: Analysis on the effectiveness of our gradient decoupling mechanism. The first line corresponds to the model before training. We train the Qwen2.5-7B-Instruct [11] model on s1K [7] dataset and evaluate on four in-domain benchmarks and 16 out-domain benchmarks, respectively. The LongCoT Response percentage is calculated on all in-domain and out-domain test data.

| Method | Gradient to Update Model Parameters | In-domain Acc@1 (avg) | Out-domain Acc@1 (avg) | LongCoT Response Percentage |
|---|---|---|---|---|
| - | - | 21.5 | 31.2 | 0% |
| SFT | - | 27.8 | 27.3 | 94% |
| DLoFT | $\mathbf{g}_{con}$ | 19.1 | 20.5 | 0% |
|  | $\mathbf{g}_{par}$ | 36.7 | 39.7 | 100% |

Table 3: Analysis on the effect of weight decay regularization. The first line corresponds to the model before training. We train the Qwen2.5-7B-Instruct [11] model on s1K [7] dataset and report the pass@1 accuracy average on four in-domain benchmarks and 16 out-domain benchmarks, respectively.

| Method | Weight Decay | In-domain Acc@1 (avg) | Out-domain Acc@1 (avg) |
|---|---|---|---|
| - | - | 21.5 | 31.2 |
| SFT | ✗ | 27.8 | 27.3 |
|  | ✔ | 26.2 | 27.4 |
| DLoFT | ✗ | 36.7 | 39.7 |
|  | ✔ | 36.0 | 39.5 |

Table 4: Efficiency analysis. The experiments are conducted on Qwen2.5-7B-Instruct [11] model with s1K [7] dataset. We report the average running time of each training step and the average GPU memory cost throughout the training.

| Method | Running Time (minutes) | GPU Memory (MB) |
|---|---|---|
| SFT | 6.2 | 69986 |
| DLoFT | 6.8 | 70684 |

general reasoning patterns and those that reflect problem-specific knowledge. As a result, the model may still internalize partial problem-specific information tied to the training data. Our findings highlight the need for more targeted training algorithms, such as our proposed DLoFT, which explicitly filters out problem-specific knowledge signals while preserving the generalizable reasoning paradigm.

## A.4 Efficiency

Compared to standard SFT, our DLoFT algorithm introduces an additional step in each training iteration, that is, the gradient decoupling step (Step 2 in Algorithm 1, where we compute a reference gradient and decouple the full gradient into the content-relevant and paradigm-relevant components via projection and orthogonalization operation. To assess the efficiency impact of this additional step, we compare our DLoFT with SFT in terms of both training speed and GPU memory cost.

As shown in Table 4, our DLoFT introduces only a minor overhead in running time per training step. This is primarily because the projection and orthogonalization operations in Step 2 are lightweight vector computations that require negligible additional compute compared to the forward and backward passes of large LLMs. Besides, the computational cost for the reference gradient $\mathbf{g}_{ref}$ is also much less than that of the full gradient $\mathbf{g}_{full}$ in Step 1, because the solution $S_i$ is generally much shorter than the thinking process $T_i$. Moreover, GPU memory usage remains comparable to that of SFT. Since DLoFT reuses existing gradients and performs additional operations in-place without requiring the

storage of large intermediate activations or duplicated models, its memory footprint remains efficient and scalable. In summary, DLoFT offers significantly improved generalization and robustness with minimal computational overhead, making it a practical replacement for SFT in real-world training scenarios.

## B LongCoT Reasoning

Long Chain-of-Thought (LongCoT) is an emerging reasoning paradigm that gained traction following the breakthrough of OpenAI o1 [1] model, which demonstrated impressive capabilities in complex reasoning tasks. Inspired by the slow thinking mechanism in humans, LongCoT encourages models to conduct a thorough thinking phase before concluding a deterministic solution. Figure 5 illustrates an example of the LongCoT reasoning paradigm.

Specifically, a LongCoT-style response consists of two sequential phases:

- A systematic *thinking* phase, where the model analyzes the problem, summarizes relevant facts, explores alternative ideas, reflects on its reasoning, and corrects potential mistakes.
- A deterministic *solution* phase, where a deterministic, step-by-step answer is provided based on the reasoning developed in the thinking phase.

## C Limitations

The high-performance, high-generalization optimization algorithm proposed in this paper for Long-CoT supervised finetuning demonstrates excellent performance across various scenarios and tasks. However, we believe that the experimental scale validated here is still limited. Real-world applications often involve much larger datasets and models (on the order of hundreds of billions). Due to limited computational resources during this work, we were unable to conduct large-scale experiments. In the future, we plan to secure additional resources to validate performance on a larger scale.

## D Impact Statement

The high-performance fine-tuning algorithm designed in this paper has many positive implications, such as enhancing the inference capabilities of existing models, allowing them to better serve a wider range of people and industries, particularly with lower data collection costs. However, there are concerns about negative impacts; for instance, this method could potentially be misused by illegal entities to train models for inhumane surveillance and authoritarian control, which legislative bodies worldwide should take seriously.

**Problem**

Every morning Aya goes for a $9$-kilometer-long walk and stops at a coffee shop afterwards. When she walks at a constant speed of $s$ kilometers per hour, the walk takes her 4 hours, including $t$ minutes spent in the coffee shop. When she walks $s+2$ kilometers per hour, the walk takes her 2 hours and 24 minutes, including $t$ minutes spent in the coffee shop. Suppose Aya walks at $s+\frac{1}{2}$ kilometers per hour. Find the number of minutes the walk takes her, including the $t$ minutes spent in the coffee shop.

**LongCoT response**

Problem Analysis    Explore New Ideas    Summarization    Self-Reflection

**Thinking**

<think>
Okay, so I need to solve this problem about Aya's walk and coffee shop stop. Let me try to parse the information step by step. …
Let me note down the given data:
1. When she walks at s km/h, the total time (walking + coffee) is 4 hours, which is 240 minutes. The coffee time is t minutes, so the walking time must be (240 - t) minutes.
2. When she walks at (s + 2) km/h, the total time is 2 hours and 24 minutes. Let me convert that to minutes as well: 2 hours is 120 minutes, plus 24 minutes is 144 minutes total. So again, the coffee time is still t minutes, so the walking time here is (144 - t) minutes.
We need to find the total time when she walks at (s + 1/2) km/h, including the t minutes. So first, maybe I need to find the value of s and t, and then compute the walking time at the new speed, add t minutes, and convert that into minutes. …
First, let's use hours because the speed is in km per hour. The distance is 9 km. …
So now we have two equations with two variables, s and t. We can solve for s and t first.
Let me write them again:
1. (9 * 60)/s + t = 240
2. (9 * 60)/(s + 2) + t = 144

…
Alternatively, 96 * 11 = 1056, so 1080 - 1056 = 24, so 11 + 24/96 = 11.25. Yep, so 11.25 = s(s + 2) …
We can discard the negative solution because speed can't be negative. So, s = ( -2 +7 ) /2 = 5/2 = 2.5 km/h
Wait, s = 2.5 km/h? Let me check if that makes sense. …
Wait, if s is 2.5, then s + 2 is 4.5 km/h. Let me check the walking times.
First scenario: 9 km at 2.5 km/h. Time in hours is 9 / 2.5 = 3.6 hours, which is 3 hours and 36 minutes (since 0.6*60=36). …
So: Walking time in hours: 9 / 3 = 3 hours. 3 hours is 180 minutes. Then add t =24 minutes, total is 204 minutes. So 204 minutes is 3 hours and 24 minutes. Hmm, that seems straightforward.
Wait, but let me check with the equations again. Maybe the problem expects something else? Let me re-read the problem.
"Find the number of minutes the walk takes her, including the t minutes spent in the coffee shop."…
So, when she walks at s + 1/2 km/h, which is 3 km/h, the total time is walking time plus t. Since t is fixed? …
Therefore, the answer should be 204 minutes, so 204.
Wait, but the problem says "the walk takes her, including the t minutes". So the "walk" here includes the coffee shop stop? So the total time is walking time plus coffee time. So yes, 180 +24=204. So 204 is the answer. Hmm, seems straightforward.
But let me check once again with the second scenario to ensure that everything is consistent. When she walks at s +2 =4.5 km/h, walking time is 9/4.5=2 hours, which is 120 minutes. Adding t=24 gives 144 minutes, which is 2h24m. That matches the problem's statement. So yes, all checks out.
Therefore, the answer is 204 minutes. Therefore, boxed answer is \boxed{204}. **Final Answer**: \boxed{204}
</think>

**Solution**

Given that Aya walks 9 kilometers and spends t minutes in the coffee shop, we need to determine the total time taken when she walks at $s + \frac{1}{2}$ kilometers per hour.
First, we set up the equations for the given scenarios:
1. When walking at \( s \) km/h, the total time is 4 hours (240 minutes):   $\frac{9}{s} \times 60 + t = 240$
2. When walking at \( s + 2 \) km/h, the total time is 2 hours and 24 minutes (144 minutes):   $\frac{9}{s + 2} \times 60 + t = 144$
Subtracting the second equation from the first to eliminate t : $\frac{9 \times 60}{s} - \frac{9 \times 60}{s + 2} = 240 - 144$
…
$ s(s + 2) = \frac{1080}{96} = 11.25 $
Solving the quadratic equation $s^2 + 2s - 11.25 = 0$ using the quadratic formula: $s = \frac{-2 \pm \sqrt{4 + 45}}{2} = \frac{-2 \pm 7}{2}$
Taking the positive root: $ s = \frac{5}{2} = 2.5 \text{ km/h} $
Substituting \( s = 2.5 \) back into the first equation to find t: $ \frac{9}{2.5} \times 60 + t = 240 $
…
$ 216 + t = 240 \implies t = 24 \text{ minutes} $
When walking at $s + \frac{1}{2} = 3$ km/h, the walking time is: $\frac{9}{3} \times 60 = 3 \times 60 = 180$ minutes
Adding the coffee shop time t :  180 + 24 = 204  minutes
Thus, the number of minutes the walk takes her, including the t minutes spent in the coffee shop, is \(\boxed{204}\).

Figure 5: Illustration of the Long Chain-of-Thought (LongCoT) style response. The problem is sampled from AIME24 [4] benchmark, and the response is generated by QwQ-32B model [22].

