# OpenReview forum: "DLoFT: Gradient-Decoupled Fine-Tuning for Generalizable Long Chain-of-Thought Reasoning"
_NeurIPS.cc/2025/Conference — NeurIPS 2025 poster_

### Official Review · Reviewer_cC52 · 2025-06-26

**Clarity:** 3
**Significance:** 2
**Originality:** 3
**Rating:** 3
**Confidence:** 4

**Summary:**

This paper aimed to improve the LongCoT’s general reasoning capabilities by decoupling the format and content gradients during training toward the LongCoT reasoning, concretely, the proposed DLoFT work first divided the format- and content-related gradients with projection and orthogonalization operations, then focused on optimizing the format-related gradients. Experimental results illustrated that the proposed fine-tuning strategy can enhance the out-of-domain LongCoT reasoning abilities, demonstrating its effectiveness in both in- and out-domain scenarios.

However, there are several key issues in this paper undermining the quality.

**Questions:**

Please refer to the above weaknesses.

**Ethical Concerns:**

["NO or VERY MINOR ethics concerns only"]

**Final Justification:**

I have read all the comments by the authors, and I will maintain my original score.

**Limitations:**

The authors didn’t discuss or mention the limitation of this work.
This work focused on the reasoning format by getting rid of the content-related information, on the one hand, such a premise was not elaborated by empirical or theoretical analysis, lacking actual motivations; on the other hand, the content information matters, especially for some question that needs a lot of specific knowledge.
There is no comparable baseline method except for the vanilla in this work, and there are numerous gradient-based research, such as [1,2,3].

[1]. Directional Gradient Projection for Robust Fine-Tuning of Foundation Models. Huang et, al. ICLR 2025.

[2]. Fast Trainable Projection for Robust Fine-Tuning. Tian et, al. NeurIPS 2023.

[3]. Improving Stability of Fine-Tuning Pretrained Language Models via Component-Wise Gradient Norm Clipping. Yang et, al. EMNLP 2023.

**Paper Formatting Concerns:**

There is no formatting issue in this paper.

**Quality:**

2

**Strengths And Weaknesses:**

**Strengths**
1. This paper conducted couples of experiments on the 7B and 32B models, and multi-domain evaluation benchmarks across various domains, results shown in tables and figures demonstrated the generalization capabilities of the proposed strategy.
2. The authors introduced the training and test data and other experimental settings in both the manuscript and the supplementary materials, and these details offered convenience for reproduction for the experiments.

**Weaknesses**
1. In the left subfigure of Figure 1, the in-domain accuracy is far lower than the out-domain, which is not in line with the current practice, implying such results were not produced by converged trained models, which undermined the explanation for the overfitting issue.
2. As claimed in Observations paragraph of Introduction and the Section 3.2, the authors assumed that it was the problem specific knowledge hindered the generalization learning, there is no empirical evidence or theoretical analysis for such an assumption.
3. The paper stated that “Focusing too much on this (specific problem-solving procedures) … limiting its ability to develop generalization reasoning skills” in Section 3.3 Step 1 paragraph, and the following statements “there is an information aimed at activating the model’s LongCoT abilities, …”. However, most scientific questions share similar reasoning patterns including derive, check, reflect, and so on, and models can also learn such format with vanilla SFT to obtain reasoning abilities for out-domains. What is the necessity of
4. In the Step 2 paragraph of Section 3.3, the authors adopted g_{ref} as the problem-specific content anchor, while there are also general reasoning formats except the problem specific contents.
5. This paper decoupled the so-called content-relevant gradient by a specific projection, simple but not explainable, the reason why such a self-defined could produce the content-relevant gradient was not interpreted. Given the calculation of the g_{con}, the extraction of g_{par} also needs to discuss, same for the fine-tuning/updating strategy.
6. The accuracy results on AIME24 seemed that the authors only conducted one-time sampling for the model inference (as there are 30 questions in AIME data), which maybe random, pass@k is a more suitable metric for LongCoT reasoning task evaluation.
7. According to the Figure 3, DLoFT for RL performed better than the vanilla SFT on the MedQA, while on the AIME24 and LiveCodeBenchv2, it seemed that it was only on par with the vanilla, which couldn’t prove the DLoFT is more effective for the cold-start setting (claimed as Section4.2.2).

---

> ### Author Response · Authors · 2025-08-01
> **Rebuttal**
>
> Thank you for your detailed and constructive comments and contribution to the academic community!
> Your feedback is of great significance to us in further improving our article. We will address each of your questions, and we hope to receive a higher rating from you if you are satisfied with our responses.
>
> ---------------------
>
> ### **Q1: Regarding the lower in-domain accuracy compared to out-domain accuracy in Figure 1**
>
> Thanks for your question. We would like to clarify this ambiguity below.
>
> - We would like to emphasize that ***all models were fine-tuned until convergence*** on their training datasets.
> - The observed ***lower in-domain accuracy does not imply under-training or lack of convergence***, but rather ***as a consequence of the intrinsic difficulty associated with the corresponding benchmark***.
>
> In this paper, "in-domain" refers to the topic on which the model was fine-tuned, and "out-domain" refers to othe topics unseen during fine-tuning. Importantly, each topic corresponds to a distinct benchmark dataset with its own difficulty and accuracy range.
> For example, in the left subfigure of Figure 1,
> - the in-domain topic is mathematical reasoning, evaluated on the well-known AIME24 benchmark, where even fully converged large language models (such as Qwen2.5 Instruct Model) typically achieve 10–30% accuracy due to the high complexity of this benchmark.
> - In contrast, the out-domain topic is medical reasoning, evaluated on the famous MedQA benchmark, where accuracy generally ranges between 40–80% for the converged open-source large language models with different sizes.
>
> To avoid potential confusion, we shall add a clarification of these benchmark-specific accuracy ranges in the revised figure caption.
>
> ---------------------
>
> ### **Q2: Regarding the assumption that problem-specific knowledge hinders generalization**
>
> We would like to clarify that our statement about "overfitting to problem-specific knowledge" isn't merely an assumption, but a conclusion drawn from *empirical observations* and *widely understood principles of machine learning*, especially in the context of data scarcity or narrow domain coverage in training data.
> - ***Standard overfitting phenomenon:*** It's a well-established fact in machine learning that when a model is trained extensively on a dataset covering only a narrow subset of the overall distribution (in our case, problem-specific knowledge compared to vast world knowledge), it tends to *overfit to the specifics of that training data*. This inevitably leads to degraded performance on unseen, out-of-distribution examples, a phenomenon often referred to as generalization gap or "catastrophic forgetting" in certain contexts.
> - ***Empirical evidence in our paper:*** We'd like to draw your attention to Figure 1 and Figure 2 in the main paper, which provides direct empirical evidence supporting our claim.
>     - Figure 1(left) clearly shows that under standard SFT, while in-domain accuracy steadily improves, the out-domain accuracy initially rises slightly then consistently declines after approximately 250 steps, eventually dropping significantly. This divergence—improving on seen domain while degrading on unseen domain—is the quintessential signature of overfitting.
>     - Furthermore, our analysis in Supplementary Material 3 ("Overfitting Problem Investigation") explicitly attributes this out-of-distribution performance degradation to "the model overfitting to problem-specific knowledge and heuristics present in the training data."
> - ***Ineffectiveness of common overfitting mitigation strategies:*** To further underscore this, we also showed that common overfitting mitigation strategies like early stopping (yellow dashed line in Figure 1) and weight decay (Section 4.3 in Supplementary Material) are ineffective.
>    - Early stopping prevents out-domain degradation but at the cost of insufficient LongCoT paradigm learning (Figure 1(right)).
>    - Weight decay doesn't improve generalization because it regularizes model parameters globally without distinguishing between gradients that encode general reasoning patterns and those that reflect problem-specific knowledge.
>    - This suggests that the overfitting isn't merely a general capacity issue, but specifically tied to the entanglement of the two learning signals: general topic-agnostic reasoning paradigm and problem-specific knowledge.
>
> In conclusion, ***our claim is not an unproven assumption*** but rather ***stems from the widely recognized overfitting phenomenon in machine learning***, and ***is strongly supported by the empirical evidence*** presented in experiments in Figure 1 \& 2, ***and the subsequent validation of DLoFT's efficacy*** in mitigating this specific problem. We will ensure this connection is made even more explicit in the revised manuscript.
>
> -----
>
> [Continued in next block]

---

> ### Author Response · Authors · 2025-08-01
> **cont.**
>
> ### **Q3: Regarding the necessity of separating problem-specific content from general reasoning patterns**
>
> - We fully agree with the reviewer that ***different scientific problems***—regardless of their surface topic—***share common reasoning patterns***, such as “derive,” “check,” “reflect,” and so on. Indeed, this point ***forms the core motivation of our work***: Since reasoning paradigms like LongCoT are inherently topic-agnostic, there is no need to rely on large-scale datasets covering a wide range of domains to learn them.
> - Motivated by this, we aim to ***teach a CoT-style model the new LongCoT reasoning paradigm using only a small amount of representative LongCoT-style data***. However, we observe that ***standard SFT fails*** in this setup.
>     - Although standard SFT can also acquire these reasoning patterns, our empirical results (Figure 1 and 2) show that it fails to achieve both learning the LongCoT reasoning paradigm and improving performance on all benchmarks with various domains.
>     - This ***isn't because the LongCoT reasoning paradigm is intransferable to other domains*** under standard SFT (actually the reasoning patterns can transfer successfully as the right subfigure of Figure 1), ***but rather because it over-memorizes specific knowledge related to the training data***, leading to performance degradation in other domains.
>     - ***Essentially, this stems from the fact that standard SFT indiscriminately learns two signals:*** (1) specific reasoning content related to the problem (e.g., the knowledge and standard procedure required for solving an eigenvalue) and (2) general reasoning paradigm irrelavant to the problem (e.g., verifying intermediate steps, backtracking upon contradiction, or brainstorming another idea).
> - After identifying this problem and analyzing its underlying reason, our DLoFT method is precisely designed to tackle this issue. By decoupling the gradients into paradigm-relevant and content-relevant components, we aim to prevent the model from memorizing narrow problem-specific details (thus preventing performance degradation in other domains) by only updating the model using the former gradient component (only relevant to the reasoning paradigm). Extensive experimental results have validated the efficacy and necessity of our decoupling mechanism.
>
> ---------------------
>
> ### **Q4: Clarification on why the general reasoning formats in $g_{\text{ref}}$ is reasonable and does not interface with DLoFT**
>
> Good question. We would like to clarify this below:
> - We agree with the reviewer that the reference gradient $g_{\text{ref}}$ contains not only problem-specific knowledge but also some general CoT-style reasoning patterns, such as step-wise instructions ("Step 1", "Step 2", etc.).
> - However, it is important to emphasize that $g_{\text{ref}}$ only contains the CoT-style reasoning patterns and does not contain any of the reasoning patterns unique to LongCoT-style thinking, such as reflection, exploring alternative ideas, or backtracking, which appear solely in the thinking part of LongCoT response.
> - Our goal is to fine-tune a model that initially only exhibits CoT-style reasoning to acquire LongCoT-specific reasoning behaviors through a small number of representative LongCoT examples.
> Since the general CoT-style patterns present in $g_{\text{ref}}$ are already part of the model's prior capabilities, they are not the reasoning behaviors we aim to learn during fine-tuning. Instead, what we seek to promote are novel and unique LongCoT-style reasoning patterns.
> - In this context, the “paradigm-relevant gradient” in our paper explicitly refers to the gradient components associated with the LongCoT reasoning paradigm only, not with CoT-style reasoning.
>
> Therefore, it is reasonable and acceptable that $g_{\text{ref}}$ includes general CoT-style reasoning formats. Their presence does not interfere with our goal, as the proposed decoupling procedure specifically targets the emergence of unique LongCoT-style behaviors. In this context, ***the inclusion of general CoT reasoning patterns in $g_{\text{ref}}$ is not a design flaw, but a reasonable, acceptable, and even desirable aspect*** in our design.
>
> We will clarify this rationale more explicitly in the revised version.

---

> ### Author Response · Authors · 2025-08-04
> **cont.**
>
> ### **Q5: Clarification on the interpretability and design principles behind our gradient decoupling mechanism**
>
> Below, we provide a detailed explanation of the rationale behind our approach, addressing your three specific questions in turn.
>
> **(1) Why we adopt only the paradigm-relevant gradient $g_{\text{par}}$ to update the model:**
>
> - As discussed in lines 38–47, although supervised fine-tuning (SFT) on LongCoT data enables the model to acquire LongCoT reasoning patterns, it also leads to a significant performance drop on out-of-domain benchmarks. Further analysis reveals that this degradation stems from the model’s tendency to overfit to problem-specific information present in the training data.
> - However, the primary goal of LongCoT fine-tuning is not to inject new knowledge into the model, but to optimize its reasoning paradigm — i.e., to improve how the model reasons with its existing knowledge, rather than what it knows. Unfortunately, the gradients obtained from SFT inherently couple two types of learning signals: (i) general problem-agnostic reasoning patterns, and (ii) problem-specific content. This coupling increases the risk of the model memorizing spurious problem-specific information present in training data during SFT, which harms generalization.
> - To address this issue, we explicitly decouple the full gradient $g_{\text{full}}$ into two orthogonal components: a content-relevant gradient $g_{\text{con}}$ and a paradigm-relevant gradient $g_{\text{par}}$. As discussed in Section 3.1, this decomposition enables us to disentangle reasoning pattern learning from content memorization. We then adopt only the paradigm-relevant gradient $g_{\text{par}}$ for model updates. In doing so, we encourage the model to improve its general reasoning capability—especially the LongCoT-specific reasoning behavior—without being biased toward any specific domain or topic seen during training.
>
> **(2) Why paradigm-relevant gradient component $g_{\text{par}}$ can be obtained by a orthogonalization operation in Eq.(4):**
>
> To decouple the two learning signals mentioned above, we decouple the full gradient $g_{\text{full}}$ into two orthogonal components:
> - content-relevant gradient component $g_{\text{con}}$ that encodes the problem-specific knowledge and heuristics
> - paradigm-relevant gradient component $g_{\text{par}}$ that captures exclusive the LongCoT reasoning pattern
>
> Formally, this orthogonal decomposition can be formulated as $g_{\text{full}} = g_{\text{con}} + g_{\text{par}}$.
> Thus, this allows us to isolate the paradigm-relevant gradient component $g_{\text{par}}$ via the orthogonalization operation $g_{\text{par}} = g_{\text{full}} - g_{\text{con}}$ as shown in Eq.(4).
>
>
> **(3) Why the projection operation in Eq.(3) yields a content-relevant gradient component $g_{\text{con}}$:**
>
> - As described in Section 3.1 and Section 3.2 (lines 166-171), the LongCoT-style response exhibits a two-phase structure that separates the "thinking process" from the "final solution". The thinking process contains both general LongCoT reasoning pattern and problem-specific information, while the "final solution" contains only problem-specific information without any LongCoT pattern.
> - Based on this structure, we compute the reference gradient $g_{\text{ref}}$ using the supervision from the "final solution" alone, as defined in Eq.(2). Since this supervision is devoid of LongCoT reasoning patterns and contains only problem-specific information, $g_{\text{ref}}$ serves as a reliable signal of content-related gradients.
> - In Eq.(3), the content-relevant gradient component $g_{\text{con}}$ is derived by projecting the full gradient $g_{\text{full}}$ onto the direction of $g_{\text{ref}}$, i.e., $g_{\text{con}} = \frac{\langle g_{\text{full}}, g_{\text{ref}} \rangle}{\| g_{\text{ref}} \|^2} \cdot g_{\text{ref}}$. Therefore, when we project $g_{\text{full}}$ (which is computed on the full LongCoT response) onto the direction of $g_{\text{ref}}$, the resulting component $g_{\text{con}}$ effectively preserves the content-relevant direction in parameter gradient space.

---

> ### Author Response · Authors · 2025-08-04
> **cont.**
>
> ### **Q6: Comparison on AIME24 with pass@k metrics**
>
> Following your suggestion, we evaluate the AIME24 results in Figure 2 and Figure 3(a). Specifically, for each question, we sample 16 different responses using temperature = 0.6 and top-p = 0.95. We then report both pass@1 (averaged on 16 responses) and pass@16 for comparison.
>
> The results are shown in Tables 1~4 below.
> It can be found that our method (DLoFT) consistently outperforms the SFT on both pass@1(avg) and pass@16 under all settings, further demonstrating the robustness of our improvements in LongCoT reasoning across both in-domain and out-domain evaluations. We shall include these results in the revised version.
>
> ---
> ***Table 1: Comparison on Qwen2.5-math-7B-Instruct finetuned on an in-domain LongCoT dataset OpenR1-Math-5K (from Figure 3(a))***
> |Method|pass@1(avg)|pass@16|
> |:-:|:-:|:-:|
> |SFT|22.5|43.3|
> |DLoFT|27.4 (+4.9)|53.3 (+10.0)|
> |SFT+RL|30.3|46.7|
> |DLoFT+RL|33.9 (+3.6)|56.7 (+10.0)|
>
> ---
> ***Table 2: Comparison on Qwen2.5-math-7B-Instruct finetuned on an out-domain LongCoT dataset OpenThoughts-Code-5K (from Figure 3(a))***
> |Method|pass@1(avg)|pass@16|
> |:-:|:-:|:-:|
> |SFT|14|26.7|
> |DLoFT|27.1 (+13.1)|50 (+23.3)|
> |SFT+RL|16|26.7|
> |DLoFT+RL|33.8 (+17.8)|53.3 (+26.6)|
>
> ---
> ***Table 3: Comparison on Qwen2.5-7B-Instruct finetuned on s1K dataset (from Figure 2(upper))***
> |Method|pass@1(avg)|pass@16|
> |:-:|:-:|:-:|
> |SFT|23.3|46.7|
> |DLoFT|26.7 (+3.4)|53.3 (+6.6)|
>
> ---
> ***Table 4: Comparison on Qwen2.5-32B-Instruct finetuned on s1K dataset (from Figure 2(bottom))***
> |Method|pass@1(avg)|pass@16|
> |:-:|:-:|:-:|
> |SFT|49.9|63.3|
> |DLoFT|60.2 (+10.3)|73.3 (+10.0)|
>
> ---
> ### **Q7: Clarification on the advantage of DLoFT as cold-start stage before RL**
>
> We suspect that your comment "DLoFT performs on par with vanilla SFT" may stem from some misunderstanding of the results in Figure 3, and we would like to clarify this point.
>
> For clarity, we have extracted the relevant results from Figure 3 and presented them in Table 5-7 below for comparison between
> "SFT+RL" and "DLoFT+RL" on AIME, LiveCodeBench, and MedQA, respectively.
> These results clearly indicate that DLoFT offers notable improvements over vanilla SFT even on AIME24 (+3.6 and +17.8) and LiveCodeBench (+1.7 and +4.2), demonstrating its advantage as a cold-start stage method before RL.
>
> ---
> ***Table 5: Comparison on AIME24 benchmark (from Figure 3(a)). Note: we use pass@1(avg on 16 responses) here***
> |Training Dataset|SFT+RL|DLoFT+RL|
> |:-|:-:|:-:|
> |OpenR1-Math-5K|30.3|33.9 (+3.6)|
> |OpenThoughts-Code-5K|16|33.8 (+17.8)|
>
> ---
> ***Table 6: Comparison on LiveCodeBench benchmark (from Figure 3(a))***
> |Training Dataset|SFT+RL|DLoFT+RL|
> |:-|:-:|:-:|
> |OpenThoughts-Code-5K|36.4|38.1 (+1.7)|
> |OpenR1-Math-5K|33.7|37.9 (+4.2)|
>
> ---
> ***Table 7: Comparison on MedQA benchmark (from Figure 3(a))***
> |Training Dataset|SFT+RL|DLoFT+RL|
> |:-|:-:|:-:|
> |Medical-o1-5K|58.9|62.8 (+3.9)|
> |OpenR1-Math-5K|43|62.5 (+19.5)|

---

> ### Author Response · Authors · 2025-08-04
> **cont.**
>
> ### **Q8 (part-1): Clarification on "The authors didn’t discuss the limitation"**
>
> We would like to clarify that we have discussed the limitations of our work in the appendix of main paper (lines 779-784).
>
>
> ---
>
> ### **Q8 (part-2): Clarification on why content information is not required in this LongCoT fine-tuning**
>
> Thank you for this insightful question, touched on a crucial distinction regarding DLoFT's application scope.
>
> We understand that this question may stem from a misunderstanding about the goal and application scope of DLoFT, so we would like to take this opportunity to clarify:
> - ***The goal of DLoFT*** is not to inject new knowledge into a model, but rather to optimize how the model reasons with existing knowledge. In particular, DLoFT is designed to refine a model’s reasoning paradigm, i.e., the manner in which it approaches problem solving. For example, in the context of Long Chain-of-Thought (LongCoT) reasoning, DLoFT enables the model to master the LongCoT reasoning paradigm — how to structure thoughts, explore ideas, reflect, and correct errors, when to reflect, when to explore different ideas, and how to iterate through a complex thinking process.
> - ***The application scope of DLoFT*** is to post-finetune an instruct model that already follows a Chain-of-Thought (CoT) reasoning paradigm—such as the Qwen-Instruct or LLaMA-Instruct series—to transform it into a model that adopts a LongCoT reasoning paradigm. These CoT-based instruct models have typically undergone pretraining (providing them with sufficient world knowledge) and supervised fine-tuning or reinforcement learning (teaching them how to apply that knowledge to solve problems, albeit within a CoT paradigm).
> - DLoFT is not designed to replace foundational knowledge injection; rather, it is intended to optimize the model’s reasoning paradigm.
>    - *If the model lacks essential foundational knowledge*, merely updating it with “reasoning-paradigm-relevant gradients” (as done in DLoFT) is insufficient to fill that knowledge gap. In such cases, continue pretraining or standard supervised fine-tuning would be more appropriate.
>    - *If the model already possesses sufficient world knowledge* and the goal is to upgrade its problem-solving paradigm from CoT to a more advanced LongCoT paradigm, then DLoFT is a more effective and efficient choice than standard SFT. It enables the transition through post-finetuning on only thousands of LongCoT-style examples, without the need to re-run the full post-training pipeline. Moreover, experimental results show that DLoFT generalizes well to out-of-domain tasks.
>
> ---
>
> ### **Q8 (part-3): Relationships with previous gradient-based regularization methods**
>
> Thanks for pointing out these works.
>
> - The mentioned works \[1,2,3] are indeed valuable contributions that propose general-purpose regularization techniques for improving robustness during fine-tuning. We appreciate this opportunity to discuss them and will include a detailed comparison and discussion in the Related Work section in our revised paper.
>
> - Importantly, these techniques are not in conflict with our method (DLoFT) and do not require an either-or choice. Instead, these regularization strategies can be compatible with our method, and can be seamlessly applied on top of the gradient component $g_{\text{par}}$ decoupled by our DLoFT algorithm.
>
> We believe integrating such general-purpose techniques with our task-specific gradient decomposition presents a promising future direction.

---

> > ### Comment · Reviewer_cC52 · 2025-08-08
> >
> > Even though the authors didn’t respond in the initial rebuttal, I would still like to respond to the discussions. I’m glad to know that the authors have adopted my suggestions for supplementing the pass@k to evaluate reasoning LLMs more comprehensively. However, there are still some concerns that have not been addressed.
> >
> > 1. The overall vanilla SFT performance of LLMs on the s1k data is great, while during my earlier investigation, the performance reported in this paper is not as significant as my experience, which is why I doubt the training convergence.
> > 2. Are there any theoretical proofs that can support the “problem-specific knowledge hinders generalization”?
> > It should also be evidenced by more qualitative and quantitative analyses. Furthermore, all the following contents were based on this assumption, without solid theoretical derivations and mechanism interpretation, the research logic lacks a critical motivation.
> > 3. In my practice, the LongCoT distillation data SFT can facilitate a broad range of reasoning tasks, for example, DeepSeek-R1-Dsitill-Qwen14B can achieve nearly 60% accuracy (from 33% or so). Such significant improvements come from the LongCT reasoning pattern R1-distilled data, given the same pretrain phase of distilled-qwen-14b and qwen-14b. And the results in the real world undermine the base assumption of this work.
> > 4. For the empirical results of vanilla SFT+RL and the proposed DLoFT+RL on different LLMs, which are largely distinct from my practice, RL (e.g., GRPO) can always improve the performance to a large margin after thousands of optimization steps, no matter the pretrained or the SFT LLMs. Such small gains of vanilla SFT+RL are unreasonable.

---

> > > ### Author Response · Authors · 2025-08-09
> > > **Response for your new comments**
> > >
> > > Dear Reviewer cC52:
> > >
> > > We sincerely thank you for your follow-up feedback and engaging in this discussion. We are glad to hear that our previous responses have addressed part of your concerns (such as pass@k evaluation). For the remaining four points, we have actively provided further clarification and explanation.
> > >
> > > As an overview:
> > > - For Q1 and Q4, we provide further explanation on the experimental results to clarify these points.
> > > - For Q2 and Q3, we offer deeper and comprehensive analysis linked to Machine Learning Theory to address the concerns.
> > >
> > > We hope these responses (in the following message blocks) can fully resolve your concerns. Meanwhile, we remain ready to respond to any further questions or feedback promptly today.
> > >
> > > Best regards,
> > >
> > > Paper 5911 Authors

---

> > > ### Author Response · Authors · 2025-08-09
> > > **For Concern-4: Clarification on the performance of "vanilla SFT+RL" in Figure 3**
> > >
> > > We would like to clarify that our reported performance of "vanilla SFT+RL" is indeed consistent with your practice.
> > >
> > > ---
> > >
> > > - **Case-1:**
> > > When performing vanilla SFT+RL on in-domain LongCoT training data, our reported performance shows clear and significant improvements compared to the performance before training—fully aligned with your experience.
> > > Specifically, recalling the results in Figure 3 of main paper (presented in Table below for your reading convenience), for three different domain LLMs, after applying vanilla SFT+RL on in-domain data, their performance significantly improves by  +13.3 on math, +2.9 on code, and +11.0 on medical, respectively. Such significant improvements are exactly consistent with your practice.
> > >
> > > | Model                      | Training LongCoT data | Before training | SFT+RL            |
> > > |----------------------------|-----------------------|------------------|-------------------|
> > > | Qwen2.5-math-7b-instruct   | math (in-domain)      | 16.7             | 30.0 (+13.3)      |
> > > | Qwen2.5-coder-7b-instruct  | code (in-domain)      | 33.5             | 36.4 (+2.9)       |
> > > | Meditron-7B                | medical (in-domain)   | 47.9             | 58.9 (+11.0)      |
> > >
> > > ---
> > >
> > > - **Case-2:**
> > > However, when applying vanilla SFT+RL on out-domain LongCoT training data, the performance after training shows no improvement or even drops compared to the model before training.
> > > This degradation primarily stems from vanilla SFT rather than RL itself. In fact, RL (e.g., GRPO) still exhibits positive improvements in these cases by recovering part of the performance drop introduced by SFT (e.g., +3.3 on math, +1.8 on code, and +2.6 on medical), which is also consistent with your practice of RL’s robustness.
> > >
> > > | Model | Training LongCoT data | Before training | SFT | SFT+RL |
> > > |----------------------------|-----------------------|------------------|-------------------|-------------------|
> > > | Qwen2.5-math-7b-instruct   | code (out-domain)      | 16.7  | 13.3 (-3.3) | 16.7 (+0.0) |
> > > | Qwen2.5-coder-7b-instruct  | math (out-domain)      |  33.5  | 31.9 (-1.6) | 33.7 (+0.2) |
> > > | Meditron-7B                | math (out-domain)  |  47.9  | 40.4 (-7.5) | 43.0 (-4.9) |
> > >
> > > ---
> > >
> > > - Building upon this, the above results further highlight that vanilla SFT is not served as a robust cold-start stage for RL. In contrast, our method (DLoFT) demonstrates strong robustness to cold-start training data: Regardless of whether the LongCoT data is in-domain or out-domain, DLoFT consistently improves the performance. For example, with in-domain data, DLoFT yields an improvement of +10.0 on math, +2.0 on code, and +9.9 on medical; with out-domain data, DLoFT achieves +10.0 gains on math, +1.8 gains on code, and +9.1 gains on medical, respectively.

---

> > > ### Author Response · Authors · 2025-08-09
> > > **For Concern-2:  Theoretical Support for "Problem-Specific Knowledge Hinders Generalization"**
> > >
> > > We would like to clarify that “problem-specific knowledge hinders generalization” is not an unfounded assumption.
> > > - Theoretically, it can be linked to the concepts of "shortcut learning" and "spurious correlations" in machine learning theory.
> > > - Empirically, the poor generalization of vanilla SFT can be found in multiple other studies.
> > > - Experimentally, our gradient decoupling mechanism provides strong evidence for it.
> > >
> > > We will clarify each of them as follows:
> > >
> > > ---
> > > **(1) Theoretical Foundations:**
> > >
> > > "Problem-specific knowledge hindering generalization" can be rigorously understood through the lens of well-established machine learning concepts such as shortcut learning and spurious correlations [1].
> > >
> > > - ***Shortcut learning*** refers to the model’s over-reliance on non-robust features that correlate statistically with training labels but do not reflect the true causal factors required for robust reasoning. These features, often called **spurious correlations**, enable models to achieve high accuracy on training but fail to transfer when faced with novel distributions. When encountering problems that require different reasoning pathways or formulas, models that have internalized these shortcuts cannot generalize effectively because they have not truly grasped the underlying first principles. Instead, they rely on memorized patterns that do not apply to new problem instances.
> > >
> > > - Within the context of LongCoT fine-tuning, the “problem-specific knowledge and heuristics” equate to these shortcut signals. For example, the SFT process may encourage the model to memorize repetitive patterns or formula usages in the training data (e.g., specific formulas, repeated solution steps), mistakenly treating them as core principles for problem-solving. Such patterns represent spurious correlations or “shortcuts.”
> > >
> > > - The paper [1] further provides a theoretical explanation for why SFT fails to generalize, attributing it to shortcut learning behaviors as described above. This aligns perfectly with our interpretation of “problem-specific knowledge” in DLoFT as a concrete manifestation of such shortcuts.
> > >
> > > ---
> > > **(2) Empirical Evidence from Other Studies:**
> > >
> > > Multiple studies have empirically demonstrated the limitations of supervised fine-tuning (SFT) in cross-domain generalization and highlighted the risk of models learning rigid, imitative behaviors instead of robust reasoning.
> > >
> > > - The paper [2] explicitly supports this viewpoint. It shows that SFT tends to memorize training data and struggles to generalize effectively in out-of-distribution (OOD) scenarios. This work clearly identifies the memorization tendency of SFT as a fundamental bottleneck limiting true generalization.
> > >
> > > - The paper [3] discusses the synergy and conflict between SFT and reinforcement learning (RL). They note that while SFT can help models learn the form of reasoning, it often locks them into rigid signals, which impedes further learning and true generalization.
> > >
> > > ---
> > > **(3) Experimental Evidence:**
> > >
> > > - Experimentally, our gradient decoupling mechanism offers strong empirical support for the assumption that problem-specific knowledge hinders generalization. Through extensive evaluations across diverse in-domain and out-of-domain LongCoT benchmarks, we observe that vanilla SFT models tend to overfit to problem-specific signals—resulting in significant performance drops when tested on out-of-domain data.
> > >
> > > - In contrast, our DLoFT method, by explicitly decoupling gradients related to reasoning paradigms from those encoding problem-specific information, consistently achieves superior generalization. It significantly improves out-of-domain performance without sacrificing in-domain accuracy, demonstrating its effectiveness in mitigating shortcut learning and memorization issues inherent to vanilla SFT.
> > >
> > > - These experimental results provide concrete evidence that isolating and reducing the influence of problem-specific knowledge during fine-tuning is crucial to achieving robust and generalizable LongCoT reasoning abilities.
> > >
> > > - Thus, our DLoFT not only remedies an inherent limitation of vanilla SFT but also opens a new pathway for more efficient and generalizable training of LongCoT reasoning ability.
> > >
> > > ---
> > > [1] Shortcut Learning of Large Language Models in Natural Language Understanding
> > >
> > > [2] SFT Memorizes, RL Generalizes: A Comparative Study of Foundation Model Post-training
> > >
> > > [3] SFT or RL? An Early Investigation into Training R1-Like Reasoning Large Vision-Language Models

---

> > > ### Author Response · Authors · 2025-08-09
> > > **For Concern-3: Clarification on the necessity of our work, given the existence of DeepSeek-R1-Distill SFT Models**
> > >
> > > Thank you for mentioning the impressive DeepSeek-R1-Distill SFT models [1]. We deeply respect their strong performance and contribution to the field.
> > > However, our work targets a ***different finetuning setup***—one that is designed to ***efficiently acquire LongCoT ability on top of already well-trained CoT-instruction-tuned models***, rather than ***restarting the entire post-training process from a base model (like DeepSeek-R1-Distill SFT models)***.
> > >
> > > ---
> > >
> > > **(1) Our Motivation for Promoting this New Setup is Threefold:**
> > >
> > > - ***Efficiency:***
> > > DeepSeek-R1-Distill models [1] and Qwen3 [2] adopt a finetuning pipeline that performing LongCoT SFT on a base model (e.g., Qwen2.5-32B-Base, Qwen3-32B-Base). This finetuning pipeline inevitably relies on a large-scale, comprehensive LongCoT dataset because such a pipeline aims to both learn the LongCoT reasoning paradigm and activate the world knowledge from pretraining. For example, DeepSeek-R1-Distill models are trained on a 800K private LongCoT dataset. This large-scale dataset requires massive data annotation cost (generating LongCoT-style responses for each problem) and extensive training resources (e.g., it takes 200 days to train on 800K LongCoT data using 8A100 GPUs.)
> > >
> > > - ***Accessibility & Generality:***
> > >     - The heavy-training setup in DeepSeek-R1-Distill models is less accessible to practitioners with limited resources. Imagine that you are a domain expert who has invested significant cost to train a domain-specific CoT instruction model (e.g., a medical expert model). If you want to add LongCoT ability, it will be prohibitively expensive to re-annotate all medical data with LongCoT responses and re-train from the base model according to the LongCoT SFT on base model setting like DeepSeek-R1-Distill models. And such a huge cost of training is just to obtain the LongCoT reasoning paradigm, not to inject any new knowledge.
> > >     - In contrast, our setup mirrors human learning. For example, to teach a new problem-solving habit, one does not need to redo all past exercises from scratch—only a small number of examples are enough to internalize the new paradigm. Imagine you're a high school student. Your previous habit is to answer questions without thinking, and without checking for errors along the way (similar to the CoT). If your teacher wants to change your habit (into a Long CoT), he simply tells you to think before you write and check your answers as you go. Then, he walks you through a few example problems using this habit. You'll quickly pick up on this new habit, without having to practice all the questions from elementary school to high school under this new habit.
> > >
> > > - ***Feasibility of “Efficient LongCoT Finetuning on a CoT Instruction Model”:***
> > >    Knowledge and reasoning paradigm are *orthogonal* properties of a model. A well-trained instruction model has already undergone extensive knowledge activation; therefore, LongCoT finetuning on such a model only needs to focus on learning the new reasoning pattern—making it possible to succeed with orders-of-magnitude less data (e.g., s1K \[3] shows strong results with only 1K curated examples).
> > >
> > > ---
> > >
> > > **(2) Why Our Method (DLoFT) is Necessary in This Setup Compared to vanilla SFT:**
> > >
> > > Previous works following this efficient LongCoT finetuning setup, such as s1K [3] and LIMO [4], have demonstrated the promise of *Efficient LongCoT finetuning on a CoT instruction model*, but they primarily train and evaluate in narrow domains (math, coding, science). Through broader experiments across 20 domains, we found:
> > >
> > > - Although vanilla SFT can improve in-domain (math, coding, science) performance, it often hurts the performance on out-domain benchmarks, violating the original goal of this efficient LongCoT finetuning setup learning without degrading prior capabilities.
> > > - We analyze and confirm from empirical experiments that this performance degradation stems from SFT’s tendency to reinforce problem-specific knowledge from fine-tuning data, which biases the model towards the distribution of these small amounts of data.
> > >
> > > Our method, DLoFT, is specifically designed to:
> > >
> > > * Emphasize learning the reasoning paradigm while minimizing the injection of problem-specific biases.
> > > * Maintain in-domain gains and preserve out-domain performance, making it a more effective and efficient choice than vanilla SFT under this setup.
> > > * Avoid re-learning knowledge that the instruction model already has—saving data, time, and compute.
> > >
> > >
> > > ---
> > >
> > > [1] DeepSeek-R1: Incentivizing Reasoning Capability in LLMs via Reinforcement Learning
> > >
> > > [2] Qwen3 Technical Report
> > >
> > > [3] s1: Simple test-time scaling
> > >
> > > [4] Less is More for Reasoning

---

> > > ### Author Response · Authors · 2025-08-09
> > >
> > > Dear Reviewer cC52,
> > >
> > > We sincerely appreciate your time and effort in reviewing our work.
> > >
> > > As the discussion phase is approaching its end within one hour, we would like to kindly inquire whether our previous responses have addressed your remaining concerns. We would be very grateful for any feedback you might have before then.
> > >
> > > If you still have any concerns, please feel free to contact us at any time — we would be more than happy to clarify further and will remain fully responsive until the end of the discussion phase.
> > >
> > > If our additional clarifications and experiments have addressed your remaining concerns, we would be sincerely grateful if you could consider a higher final score judgment. Thank you very much for your consideration.
> > >
> > > Best regards,
> > >
> > > Paper 5911 Authors

---

> ### Author Response · Authors · 2025-08-07
>
> Dear Reviewer cC52:
>
> Thank you very much for your time and effort during the initial review period.
>
> We have done our best to address your comments and questions in our rebuttal. As the discussion phase is nearing its end, we would greatly appreciate it if you could let us know whether our responses have sufficiently addressed your concerns. Of course, we are also more than happy to further clarify or discuss any remaining issues you may have.
>
> We look forward to hearing from you, and thank you again.
>
> Best regards,
>
> Paper 5911 Authors

---

> ### Author Response · Authors · 2025-08-09
> **For Concern-1: Clarification on the Full Convergence on s1K. (Consistency with the official results in s1K paper confirms full convergence)**
>
> Thanks for the careful examination of our results. We have re-verified our training process and confirm that the models were fully converged; the performance reported in our paper is correct and consistent with the original results in s1K paper.
>
> To further clarify, we directly compared "the vanilla SFT results reported in our paper" with that "reported in original s1K paper [1]".
>
> - ***Experimental setup:***
> The s1K paper reports performance of the Qwen2.5-32B-Instruct model fine-tuned on s1K data, which exactly matches the setup in Table 2 (bottom) of our paper. So, we compare under this experimental setup.
>
> - ***Results from the s1K paper:***
> As Table 1 of the original paper, the AIME24 pass@1 performance is 50.0.
> Note that "s1 w/o BF" in the second-to-last row is the correct baseline to compare against, rather than the "s1-32B" in the last row, because "s1-32B" uses a budget-forcing sampling strategy rather than standard sampling.
>
> - ***Our reported results:***
> Following your earlier suggestion, we reported pass@1 averaged over 16 sampling runs to reduce randomness, and updated the performance in Table 2 (bottom) with Table 4 from the response for Q6 in previous messages.
> For easy reading, we pasted Table 4 from the Q6 response below.
> As you can see, our reported AIME24 pass@1 performance is 49.9, which is only 0.1 lower than the 50.0 reported in the original s1K paper—well within the range of random variation.
>
> ---
> ***Table 4 (from the response for Q6 above): Comparison on Qwen2.5-32B-Instruct finetuned on s1K dataset (from Figure 2(bottom))***
> |Method|pass@1(avg)|pass@16|
> |:-:|:-:|:-:|
> |SFT|49.9|63.3|
> |DLoFT|60.2 (+10.3)|73.3 (+10.0)|
>
> ----
>
> In summary, the pass@1 performance in the s1K paper is 50.0, and our reported performance achieves 49.9 — the negligible 0.1 difference confirms that our training is correct and fully converged. Furthermore, applying our DLoFT under the same setup improves pass@1 to 60.2, a remarkable +10 point gain over vanilla SFT, showing the effectiveness of our method.
>
> ---
>
> [1] s1: Simple test-time scaling

---

### Official Review · Reviewer_bbCD · 2025-06-30

**Clarity:** 1
**Significance:** 3
**Originality:** 3
**Rating:** 4
**Confidence:** 3

**Summary:**

The paper introduces DLoFT, an algorithm designed to enhance the generalizability of LongCoT reasoning in large language models while mitigating overfitting to problem-specific knowledge. The authors claim that DLoFT significantly improves the generalization behavior of LongCoT abilities compared to SFT while maintaining strong in-distribution performance.

**Questions:**

Please see Weaknesses. My primary focus lies in points 1 through 7.

**Ethical Concerns:**

["NO or VERY MINOR ethics concerns only"]

**Final Justification:**

I appreciate the additional clarifications, and as a result, I have raised my evaluation.

**Limitations:**

yes, some Limitations are discussed in Appendix Limitations.

**Paper Formatting Concerns:**

This paper demonstrates proper formatting throughout.

**Quality:**

2

**Strengths And Weaknesses:**

> **Strengths:**
>
> (1) Good visualizations and typesettings.
>
> (2) A well-justified research problem: models trained in this way (LongCoT SFT) tend to overfit problem-specific knowledge and heuristics, leading to degraded out-of-distribution performance. Authors try to concentrate on internalizing the LongCoT paradigm rather than memorizing problem-specific knowledge and heuristics.
>
> (3) Well-founded starting point: Gradient decoupling.

> **Weaknesses:**
>
> (1) Line 64:  "the model is updated using only the paradigm-relevant gradient component". If the foundational knowledge of the base model requires updating, merely processing paradigm-relevant gradients would be insufficient.
>
> (2) Lines 183-184: "For each training example in the mini-batch, represented as a tuple (Pi ,Ti ,Si ) , where Pi denotes the problem, Ti signifies the exploratory thinking process, and Si indicates the deterministic solution". The paper does not explicitly specify the method for distinguishing between Ti (thinking phase) and Si (solution phase) in the main text - is the \<think\> tag used as the delimiter?
>
> (3) In your Figure 1 in supplementary material. The example between Si and Ti shown in Appendix Figure 1 indicates these components are not fully decoupled, suggesting the need for more refined differentiation criteria between the solution and thinking phases.
>
> (4) Equations 1 and 2 do not provide clarification on the variable $\mathcal{B}$, although it can be understood to represent the batch size.
>
> (5) The linear dependence between $g_{con}$ and $g_{ref}$ (Equation 3) raises the question: What would be the impact of directly substituting $g_{con}$ with $g_{ref}$?
>
> (6) Lines 215-216, "By **projecting $g_{full}$** onto $g_{ref}$, we isolate the component of $g_{full}$ that pertains exclusively to problem-specific reasoning."  However, in your Equation 3, you try to use $g_{con}= \lambda (g_{full},g_{ref})\times g_{ref}$. **This appears to project $g_{ref}$ rather than $g_{full}$**.
>
> (7) Line 244 "we adopt the Qwen2.5-Instruct series". [1] shows that it is easy to get significant performance gains on Qwen models even with completely spurious reward signals. The experimental validation conducted solely on the Qwen2.5 model series lacks sufficient persuasiveness.
>
> [1] Spurious Rewards: Rethinking Training Signals in RLVR. arXiv:2506.10947.
>
> (8) Line 563, the author say "The code and data will be made publicly available upon acceptance through peer review. " The extent of reproducibility cannot be fully determined based on the current information.

---

> ### Author Rebuttal · Authors · 2025-07-31
>
> Thank you for your detailed and constructive comments and contribution to the academic community!
> Your feedback is of great significance to us in further improving our article. We will address each of your questions, and we hope to receive a higher rating from you if you are satisfied with our responses.
>
> ---------------------
>
> ### **Q1: On the sufficiency of paradigm-relevant gradients when foundational knowledge may be lacking**
>
> Thank you for this insightful question, touched on a crucial distinction regarding DLoFT's application scope.
>
> ***“Knowledge injection”*** and ***“learning reasoning paradigms”*** are two complementary rather than mutually exclusive learning objectives.
> ***DLoFT is not designed to replace foundational knowledge injection; rather, it is intended to optimize the model’s reasoning paradigm and is thus complementary to knowledge updating techniques.***
> - *If the model lacks essential foundational knowledge*, merely updating it with “reasoning-paradigm-relevant gradients” (as done in DLoFT) is insufficient to fill that knowledge gap. In such cases, continue pretraining or standard supervised fine-tuning would be more appropriate.
> - *If the model already possesses sufficient world knowledge* and the goal is to upgrade its problem-solving paradigm from CoT to a more advanced LongCoT paradigm, then DLoFT is a more effective and efficient choice than standard SFT. It enables the transition through post-finetuning on only thousands of LongCoT-style examples, without the need to re-run the full post-training pipeline. Moreover, experimental results show that DLoFT generalizes well to out-of-domain tasks.
>
> We understand that this question may stem from a misunderstanding about the goal and application scope of DLoFT, so we would like to take this opportunity to clarify:
> - ***The goal of DLoFT*** is not to inject new knowledge into a model, but rather to optimize how the model reasons with existing knowledge. In particular, DLoFT is designed to refine a model’s reasoning paradigm, i.e., the manner in which it approaches problem solving. For example, in the context of Long Chain-of-Thought (LongCoT) reasoning, DLoFT enables the model to master the LongCoT reasoning paradigm — how to structure thoughts, explore ideas, reflect, and correct errors, when to reflect, when to explore different ideas, and how to iterate through a complex thinking process.
> - ***The application scope of DLoFT*** is to post-finetune an instruct model that already follows a Chain-of-Thought (CoT) reasoning paradigm—such as the Qwen-Instruct or LLaMA-Instruct series—to transform it into a model that adopts a LongCoT reasoning paradigm. These CoT-based instruct models have typically undergone pretraining (providing them with sufficient world knowledge) and supervised fine-tuning or reinforcement learning (teaching them how to apply that knowledge to solve problems, albeit within a CoT paradigm).
>
> ---------------------
>
> ### **Q2: Clarification on how $T_i$ and $S_i$ are separated**
>
> Yes, as a common convention in LongCoT-style datasets, each full response follows a standardized format:
>
> \<think\> THINKING PROCESS \<\/think\> SOLUTION
>
> The special tag "\<\/think\>" serves as the delimiter that separates the exploratory thinking process ($T_i$) from the final deterministic solution ($S_i$).
> Given this structure, one can reliably split any LongCoT response using the "\<\/think\>" tag.
> For example, a Python snippet for parsing $T_i$ and $S_i$ would be:
> ```
> T_i, S_i = full_response.split("</think>")
> T_i = T_i.replace("<think>", "").strip(" \n")
> S_i = S_i.strip(" \n")
> ```
> To further aid understanding, we have illustrated an example of such a LongCoT-style response in Figure 1 of the Supplementary Material, where the structure and separation of $T_i$ and $S_i$ are clearly shown.
>
> ---------------------
>
> ### **Q3: Clarification on the semantic and structural decoupling between $T_i$ and $S_i$ in LongCoT response**
>
> We appreciate the reviewer’s careful inspection of the example shown in Appendix Figure 1. The concern seems to be whether the thinking phase ($T_i$) and solution phase ($S_i$) are fully decoupled — that is, whether the boundary between the two is both structurally well-defined and semantically meaningful.
>
> - ***Structural Decoupling.***
> In LongCoT data, the thinking and solution phases follow a canonical structure, with a special delimiter "\<\/think\>" that separates the two parts (as demonstrated in the above response for Q2). Importantly, this "\<\/think\>" delimiter is now widely adopted in recent deep reasoning models, such as DeepSeek-R1, as a standard practice for separating thinking from final solution. The example shown in Figure 1 of the Supplementary Material adheres to this convention.
>
> - ***Semantic Decoupling.***
> Beyond structural decoupling, LongCoT also promotes a clear functional division:
>     - The thinking phase ($T_i$) involves open-ended reasoning: intermediate steps, scratch work, alternative hypotheses, checks, and so on. It is inherently non-deterministic and may involve trial-and-error or backtracking.
>     - The solution phase ($S_i$), by contrast, serves as a concise and deterministic answer, usually synthesized based on the preceding reasoning steps. It is expected to be directly usable for evaluation (e.g., exact-match or numerical comparison).
>
>     This semantic distinction aligns with how humans distinguish between "working through a problem" and "writing down the final answer".
>
> ---------------------
>
> ### **Q4: Clarification on Variable $\mathcal{B}$**
>
> We thank the reviewer for pointing this out. The variable $\mathcal{B}$ in Eq.(1) and Eq.(2) indeed denotes the mini-batch size. We will revise the manuscript to explicitly clarify this notation.
>
>
> ---------------------
>
> ### **Q5: The impact of directly substituting $g_{\text{con}}$ with $g_{\text{ref}}$**
>
> Good question. Indeed, we have discussed this question in Supplementary Material Section 4.1, where we experimentally and empirically investigate whether the projection step in Eq.(3) is necessary, that aligns with your question: the impact of replacing $g_{\text{con}}$ with $g_{\text{ref}}$.
>
> Specifically, we compare two variants:
> - using the projection operation as defined in Eq.(3) of the main paper, that is $g_{\text{con}} = \frac{\langle g_{\text{full}}, \\ g_{\text{ref}} \rangle}{\|| g_{\text{ref}} \||^2} \cdot g_{\text{ref}}$
> - skipping the projection and directly assigning $g_{\text{con}} = g_{\text{ref}}$
>
> For convenience, we put the results in Table 1 of the Supplementary Material in the table below.
>
> | Method | Projection | In-domain Acc@1 (avg) | Out-domain Acc@1 (avg) |
> |:------:|:----------:|:---------------------:|:----------------------:|
> | -      | -          | 21.5                   | 31.2                    |
> | SFT    | -          | 27.8                   | 27.3                    |
> | DLoFT  | ✗          | 30.3                   | 31.8                    |
> | DLoFT       | ✓          | 36.7                   | 39.7                    |
>
> The results clearly demonstrate that:
> - The variant with projection operation yields better performance than its direct substitution counterpart ($g_{\text{con}} = g_{\text{ref}}$) on both in-domain and out-domain generalization. This highlights that the projection step plays a critical role in more precisely isolating problem-specific gradients, and its removal leads to incomplete decoupling, thereby reducing the effectiveness of the filtering problem-specific knowledge and heuristics.
> - Omitting the projection still achieves better performance than standard SFT and partially mitigates the out-of-domain performance drop.
>
> To ensure this clarification is more visible, we will move the relevant analysis from the supplementary material into the main paper.
>
> ---------------------
>
> ### **Q6: Clarification on the projection operation in Eq.(3)**
>
> We would like to clarify the meaning of Eq.(3) to address the possible misunderstanding.
>
> The projection formula we use is: $g_{\text{con}} = \frac{\langle g_{\text{full}}, \\ g_{\text{ref}} \rangle}{\|| g_{\text{ref}} \||^2} \cdot g_{\text{ref}}$.
> - This expression calculates the component of the full gradient $g_{\text{full}}$ along the direction of $g_{\text{ref}}$. More concretely, it finds the scalar projection coefficient $\lambda = \frac{\langle g_{\text{full}}, g_{\text{ref}} \rangle}{\| g_{\text{ref}} \|^2}$, quantifies how much of $g_{\text{full}}$ lies in the direction of $g_{\text{ref}}$, and then scales $g_{\text{ref}}$ by this amount.
> - Therefore, ***$g_{\text{con}}$ is indeed the projection of $g_{\text{full}}$ onto the vector $g_{\text{ref}}$***, isolating the component of $g_{\text{full}}$ that corresponds to the problem-specific reasoning gradient direction $g_{\text{ref}}$.
> It is not projecting $g_{\text{ref}}$ itself; rather, $g_{\text{ref}}$ serves as the direction basis for the projection of $g_{\text{full}}$.
>
> This is a standard vector projection operation in linear algebra, derived from the geometric interpretation of the dot product as follows:
> - Recall that the dot product between two vectors $a$ and $b$ is: $ \langle a, b \rangle = \|| a \|| \cdot \|| b \|| \cdot \cos \theta$, where $\theta$ is the angle between the two vectors.
> - Rearranging this gives: $\|| a \|| \cos \theta = \frac{\langle a, b \rangle}{\|| b \||}$, which is the length of the projection of $a$ onto the unit vector in direction $b$.
> - If we want the full vector form of this projection, we multiply this scalar by the unit vector in the direction of $b$, i.e., $\frac{b}{\|| b \||}$, leading to: $\text{proj}_{b}(a) = \frac{\langle a, b \rangle}{\|| b \||} \cdot \frac{b}{\|| b \||} = \frac{\langle a, b \rangle}{\|| b \||^2} \cdot b$
>
> We hope this clarifies the misunderstanding.
>
> [Continued in next block]

---

> > ### Comment · Reviewer_bbCD · 2025-08-05
> >
> > Thank you for the clarifications and additional experiments! Having considered all reviews and responses to them, I see no important weakness.

---

> > > ### Author Response · Authors · 2025-08-05
> > > **Thanks for the response.**
> > >
> > > Thank you for your response! We are more than happy to address all of your concerns. If you are satisfied with our reply, we would greatly appreciate a higher score from you.

---

> ### Author Response · Authors · 2025-08-04
> **cont.**
>
> ### **Q7: Comparison on beyond Qwen models: Demonstrating our effectiveness on LLaMA model**
>
> To address this concern, we have added new experiments using a completely different model series: LLaMA.
> Specifically, we fine-tuned LLaMA3.1-8B-Instruct on the s1K dataset under the same experimental setting as described in Section 4.1, and evaluated on a variety of benchmarks for 20 domains.
>
> The results, presented in the Table below, demonstrate that:
> - Finetuning LLaMA model using standard SFT still leads to performance degradation in some out-domain benchmarks.
> - Our method (DLoFT) continues to significantly outperform both the SFT and the original model across all benchmarks, confirming that the observed gains are not artifacts of the Qwen model series and are instead indicative of improved generalization.
>
> These results strengthen the conclusion that DLoFT leads to more generalizable reasoning capabilities across domains, and its effectiveness is not specific to a particular model architecture such as Qwen.
>
>
> | Method | math    | physics   | chemistry | biology | code    | medicine | computer | electronic | communication | mechanical | astronomy | geography | civil  | agriculture | economics | history | law    | literature | philosophy | sociology |
> |--------|:-------:|:---------:|:---------:|:-------:|:-------:|:--------:|:--------:|:----------:|:-------------:|:----------:|:---------:|:---------:|:------:|:-----------:|:---------:|:-------:|:------:|:----------:|:----------:|:----------:|
> | -      | 6.7     | 13.2      | 12.7      | 21.6    | 17.3    | 62.6     | 23.5     | 15.9       | 20.6          | 21.6       | 15.1      | 32.3      | 13.7   | 15.1        | 24.4      | 20      | 26.7   | 20.9       | 27.7       | 31.5       |
> | SFT    | 10.4 (+3.7) | 16.8 (+3.6) | 15.2 (+2.5) | 26.5 (+4.9) | 15 (-2.3) | 55.8 (-6.8) | 26.3 (+2.8) | 15 (-0.9) | 17.5 (-3.1) | 18.4 (-3.2) | 10.6 (-4.5) | 29.5 (-2.8) | 15 (+1.3) | 9.6 (-5.5) | 26.6 (+2.2) | 15.4 (-4.6) | 28 (+1.3) | 18 (-2.9) | 21.5 (-6.2) | 32.4 (+0.9) |
> | DLoFT  | 13.7 (+7.0) | 26.2 (+13.0) | 20.5 (+7.8) | 32.0 (+10.4) | 18.9 (+1.6) | 68 (+5.4) | 39.8 (+16.3) | 20.2 (+4.3) | 21.7 (+1.1) | 25.8 (+4.2) | 17.8 (+2.7) | 35.1 (+2.8) | 16.3 (+2.6) | 15.8 (+0.7) | 30.0 (+5.6) | 22.4 (+2.4) | 35.6 (+8.9) | 30.4 (+9.5) | 36.7 (+9.0) | 35.7 (+4.2) |
>
> ---
>
> ### **Q8: Reproducibility Commitment and Clarifications**
>
> We are fully committed to open source: we shall release all code, data, and model checkpoints upon acceptance to ensure the community can easily reproduce and build upon our work.
>
> In addition, our paper already provides sufficient details to support reproducibility, as outlined below:
> - Experimental Setup: Section 4.1 and Supplementary Material Section 2 provide comprehensive details about training configurations, datasets used, and evaluation protocols.
> - Algorithmic Details: We present the full algorithmic procedure in Algorithm 1. Notably, our method is very easy to implement on top of standard SFT. It only introduces three simple additional steps:
>
>     (1) One forward and backward pass, as shown in Eq. (2);
>
>     (2) A matrix projection operation, as shown in Eq. (3);
>
>     (3) A matrix subtraction operation, as shown in Eq. (4).
>
> We hope this clarifies that our method is not only reproducible but also implementation-friendly.

---

### Official Review · Reviewer_Dhab · 2025-07-02

**Clarity:** 3
**Significance:** 3
**Originality:** 4
**Rating:** 5
**Confidence:** 4

**Summary:**

This paper proposes to improve the initiation phase for long chain of thought (longCoT) reasoning, in which the model is fine tuned on some longCoT examples, by separating the gradient of this fine tuning into two components - a "paradigm" component and a "solution" component - and then fine tuning only from the "paradigm" component. This is done by computing the full gradient of the longCoT sample, computing its projection onto a gradient representing the 'solution' only, and then removing that projection from the full gradient (the remaining gradient is assumed to represent the longCoT paradigm: exploration of multiple different approaches, reflection on approaches tried, etc). The gradient representing the solution only is taken by breaking the same longCoT sample into its problem (P), exploratory thinking (T), and solution (S) segments, and backpropagating from the loss of only S as a direct continuation from P, i.e. Loss(S|P) (without T). The approach is demonstrated to have superior performance to initation via direct fine tuning on longCoT, for both in- and out- of distribution (relative to the fine tuning) problem domains, and especially for OOD.

**Questions:**

the issues i have are raised in the strengths/weaknesses. these are additional comments for improving presentation
- lines 267-269 feel contradictory with 156-157
- line 28 should be backed by some citations showing 'community' (academic community?) has been trying
- line 33 'proprietary' maybe you mean 'ownership'?
- it should be either 'our method' or 'DLOFT', not 'our DLOFT' (i suggest 'DLOFT')
- lines 70-71 should reference the relevant tables for these results
- lines 110-111 i do not see where in your experiments you show that SFT has narrow generalisation specifically due to overfitting, this seems to be an assumption rather than an investigation
- lines 124-125 - improve phrasing
- 125-127 clarify - is this what the just-mentioned papers found?
- 135-138 space waste, similar for 236-240
- 189-196 example of information that has already been repeated several times at this point
- 223-227 again repetition of same info

**Ethical Concerns:**

["NO or VERY MINOR ethics concerns only"]

**Final Justification:**

already liked the work, and hoping presentation will improve as suggested

**Limitations:**

yes

**Quality:**

3

**Strengths And Weaknesses:**

strengths
- I rather like the approach - I find it straightforward, intuitive, and elegant.
- The results also seem to show strong positive impact from the approach
- Some nice figures, esp. fig 2 has quite clear take-home message


weaknesses
- The writing/presentation needs improvement (more comments in "questions" section):

-- The paper feels very repetitive - it feels like almost nothing is said given the space used, and a lot of space is spent constantly on retelling me that the full loss in fact represents two things, or on other space wasting things (e.g. explaining structure of an upcoming section). It feels almost empty as a result, as once this idea is understood there is not much else to learn from reading. Meanwhile several details or results are deferred to the appendix. I think a rewrite would allow fitting these into the main paper.

-- Figure 1 has no clear discussion, the narrative just throws a reference to it without details, its hard to make much of it in this way. Presumably the same paragraph referencing it (lines 38-47) is discussing what is seen in it, but it is not really tied properly to the figure, its hard to understand how to relate the two. The left subplot seems to show out-of-domain performance is better than in- for both SFT and DLOFT, surely this merits discussion.

-- The caption of fig 1 also doesnt quite align with what is in it - especially the statement that OOD performance starts to degrade continuously after the yellow line, in contrast with what is seen in the fig.

-- (There are also various grammar/English oddities, the paper really needs a proper reread and polish in this regard.)

- Some design choices odd: in particular the choice to sample 5k samples from the single domain datasets, when many more are available and the models are fine tuned for 10 epochs on these samples. This is not properly motivated.

---

> ### Author Response · Authors · 2025-08-01
> **Rebuttal**
>
> Thank you for your thoughtful and constructive feedback. We are encouraged by your recognition of the motivation, empirical findings, and broader impact of our work. We appreciate your time and effort in providing detailed comments, which have helped us improve both the clarity and quality of our paper. Below, we respond to each point in detail.
>
> ---
>
> ### **Q1: Response to the detailed suggestions for writing/presentation improvement**
>
> We sincerely thank the reviewer for the constructive feedback on the writing and presentation. We have carefully addressed each of the specific comments mentioned in the "questions" section and shall make the corresponding revisions in the revision.
> Below, we provide clarifications and actions taken for each point:
>
> ------
>
> **(1) Lines 267-269 seem contradictory with lines 156-157**
>
> First, review the statements in these lines:
> - line 156-157: Typically, models require more training iterations to internalize strong and stable LongCoT abilities due to the complexity of the LongCoT reasoning process.
> - line 267-269: Prior studies [37, 20] have revealed that learning LongCoT reasoning capabilities does not require hundreds of thousands of data, because it focuses on changing the reasoning paradigm rather than memorizing a large amount of knowledge.
>
> Then, we would like to clarify that these two statements may seem contradictory at first glance, but they refer to different aspects of the learning process.
> - Line 156–157 emphasize that LongCoT reasoning is inherently more complex and thereby requires more training iterations to be effectively learned.
> - In contrast, lines 267–269 refer to the amount of training data required for learning LongCoT pattern. Regarding this point, our view is that the goal of LongCoT finetuning is to learn a reasoning paradigm that is general and transferable to different problems, rather than to inject knowledge. Therefore, it only requires a few representative data with LongCoT responses, rather than millions of training data. This aligns with prior studies [37, 20].
>
> In short, while learning LongCoT requires more iterations per sample to converge due to its complexity, it does not require a large amount of training data. We shall revise the paper to make this distinction clearer. Thanks for your question.
>
> ------------
>
> **(2) The meaning of “community” in Line 28 and add relevant citations**
>
> Thank you for the suggestion. We clarify that by "community," we refer to both the academic and industrial communities. We will revise the wording to make this explicit and will also add appropriate citations to support the claim. Specifically, we will cite the following relevant works [19, 20, 27, 25, 38] and [16, 8, 33] for academic community and industrial community, respectively.
>
> ------------
>
> **(3) "proprietary" vs. "ownership" in line 33**
>
> We appreciate the reviewer’s suggestion regarding the term choice. To clarify, the term “proprietary” emphasizes that certain datasets in some fields are restricted and confidential, often due to privacy, legal, or commercial reasons. For example, user data in healthcare or finance domains is typically proprietary. This differs from “ownership,” which mainly refers to who legally holds the rights to the data, whereas “proprietary” highlights limited access and usage restrictions.
>
> ------------
>
> **(4) It should be either "our method" or "DLoFT", not "our DLoFT"**
>
> Thank you for pointing it out. We will revise the manuscript to consistently use either "our method" or "DLoFT," avoiding the phrase "our DLoFT."
>
>
> ------------
>
> **(5) Add reference for results in lines 70-71**
>
> We agree that explicitly referencing the relevant figures and sections will make the connections clearer and improve the overall readability. We will update the manuscript to include a direct reference to Figure 2 for lines 70-71, and Figure 3 for lines 72-73.
>
>
> ------------
>
> **(6) Clarification on SFT’s Narrow Generalization and Overfitting in line 110-111**
>
> We would like to clarify that the statement in lines 110-111 is not merely an assumption but is grounded in our empirical observations. Specifically, as discussed in the “observations” paragraph (lines 38-47) and detailed further in Supplementary Material Section 3, our experiments demonstrate that standard SFT trained on LongCoT data with narrow topical coverage leads to poor generalization (out-domain performance degradation) on diverse novel topics. This degradation is linked to overfitting on problem-specific knowledge contained in the training data.
>
> ------------
>
> **(7) Improve phrasing for lines 124-125**
>
> Thank you for the suggestion. We therefore revise as follows:
> Although DeepSeek-R1-Zero [8] has achieved notable success, recent studies [38] have observed that Zero-RL does not consistently induce the LongCoT reasoning paradigm, as evidenced by the lack of stable increase in response length and complexity during training.
>
> ---
> [Continued in next block]

---

> > ### Comment · Reviewer_Dhab · 2025-08-06
> >
> > thanks, do please make an appropriate update to lines 110-111 in the revised manuscript too (ie connect these to the relevant results, as you do here)

---

> ### Author Response · Authors · 2025-08-01
> **cont.**
>
> **(8) Source of the statement in lines 125-127**
>
> Thank you for pointing this out. To clarify, the statement in lines 125-127 — that a supervised fine-tuning (SFT) cold-start phase on LongCoT-style data is essential to instill the LongCoT reasoning paradigm before applying reinforcement learning (RL) for further enhancement — is indeed a conclusion supported by the recent study [38] mentioned just prior.
> We will revise the manuscript to make this connection clearer and explicitly attribute this insight to [38] for better clarity.
>
> ------------
>
> **(9) Space waste for line 135-138 and 236-240**
>
> We agree with the reviewer that these summary-style transitions introduce unnecessary space consumption.
> We will remove these lines and streamline the section transitions to improve conciseness.
>
> ------------
>
> **(10) Repetition for lines 189-196 and 223–227**
>
> Thank you for pointing this out.
> We agree that this paragraph partly reiterates previously explained concepts, particularly the distinction between the content-relevant and paradigm-relevant gradient components.
> In the revised version, we will streamline this explanation and avoid repeating similar contents.
>
> ------------
>
> ### **Q2: Repetitiveness of main paper, and deferral of details or results to the supplementary material**
>
> Thank you for your constructive feedback of great significance to us in further improving our paper.
>
> - We agree that the manuscript currently includes excessive repetition around the central idea that the full gradient/loss contains two entangled components. In the revised version, we will significantly reduce such redundancy and rewrite the relevant sections to improve conciseness and clarity.
> - In addition, we will restructure the paper to bring key analyses and results from the supplementary material—such as the analysis in Section 3, the ablation studies and the efficiency analysis in Section 4 —into the main body of the paper. This will enhance the information density and ensure that readers can access all important insights without needing to refer to the supplementary material.
>
> ------------
>
> ### **Q3: More explanation for Figure 1**
>
> We appreciate your suggestion. Indeed, a detailed explanation for Figure 1 is initially provided in Supplementary Material Section 3. In the revised version, we shall move this explanation to the main paper to ensure Figure 1 is well contextualized and clearly interpreted.
>
> (1) Here, we provide some explanation for what Figure 1 conveys:
> - As shown in Figure 1, although the accuracy on the in-domain benchmark steadily improves during training, the accuracy on the out-domain benchmark initially increases slightly (from 55.3 to 57.0) and then begins to consistently decline after 250 step, eventually dropping to 46.9. This trend suggests that the out-of-distribution performance degradation is due to the model overfitting to problem-specific knowledge and heuristics present in the training data.
> - In addition, common overfitting mitigation strategy, such as early stopping (stopping training before out-domain performance degradation), has been shown to be ineffective. Although simply applying early stopping can prevent a performance degradation on out-domain benchmark, it leads to both the limited performance gains and insufficient learning on the LongCoT reasoning paradigm. As shown in Figure 1(left), if training is stopped early at step 250 (indicated by the yellow dashed line), we forfeit approximately 7 points on the accuracy of in-domain benchmark. Simultaneously, as Figure 1(right), only 3.3% of the in-domain test examples exhibit LongCoT behavior at the early stopping moment, indicating that the model has not yet sufficiently internalized the LongCoT reasoning paradigm.
> - These observations underscore a key challenge: naive supervised fine-tuning (SFT) is prone to overfit to problem-specific knowledge and heuristics present in the training data, rather than internalizing a generalizable LongCoT reasoning paradigm.
> In contrast, our DLoFT not only ensures a steady increase in in-domain and out-domain performance (as Figure 1(left)), but also demonstrates faster learning of the LongCoT paradigm (see Figure 1(right)).
>
> (2) Additionally, we would like to address your concern regarding the lower in-domain accuracy compared to out-domain accuracy in Figure 1.
> - The phenomenon “in-domain accuracy is lower than out-domain accuracy" does not imply undertraining or lack of convergence, but rather is a consequence of the intrinsic difficulty associated with the corresponding benchmark.
> - For example, in Figure 1(left), the in-domain (math) benchmark AIME24 is very challenging, where LLMs like Qwen2.5-7B level model typically achieve 10–30% accuracy. In contrast, the out-domain (medical) benchmark is MedQA, where accuracy generally ranges between 40–80% for Qwen2.5-7B level model.
>
> To avoid potential confusion, we shall add a clarification of these benchmark-specific accuracy ranges in the revised paper.

---

> ### Author Response · Authors · 2025-08-01
> **cont.**
>
> ### **Q4: Clarification on Figure 1 caption for out-domain performance trend**
>
> The statement in the caption—"starts to degrade continuously after the yellow line"—was intended to describe the overall downward trend of out-domain performance after step 250. However, we agree that the phrasing "degrade continuously" may have inadvertently suggested a perfectly monotonic decrease, which is inappropriate. In reality, while there are minor fluctuations, the general trajectory of the out-domain accuracy clearly trends downward and does not recover to its previous peak.
>
> To resolve this ambiguity, we will revise the caption to more accurately reflect the observed behavior. The updated sentence will be: "... the out-of-domain accuracy begins to exhibit a downward trend after the early stopping point (yellow line), with no subsequent recovery beyond the peak observed before step 250."
> This phrasing better aligns with the data shown in the figure and avoids the implication of a strictly monotonic decline.
>
> --------
>
> ### **Q5: Reason for using 5K sample during training**
>
> In our main experiments, we used a 5K-sample subset randomly drawn from the full LongCoT dataset.
> We chose to use 5K samples per domain as an empirically effective trade-off between performance and computational cost.
> In our early experiments, we observed that increasing the number of training samples beyond 5K only led to marginal performance gains while incurring significantly higher training cost.
> To justify this design, we now include these experimental results (Tables 1–3 below) that systematically compare model performance at different dataset sizes: 2.5K, 5K, 7.5K, 10K, and 20K. These experiments are conducted and evaluated across three domains: medical, coding, and math reasoning, using three different excellent instruction-tuned domain-specific models, respectively.
>
> These results clearly show that:
> - DLoFT demonstrates strong data efficiency and has a low data volume requirement. The performance improves noticeably when increasing from 2.5K to 5K, but plateaus or only slightly improves beyond 5K. This indicates that 5K is a data level at which performance and efficiency achieve a trade-off.
> - In contrast, standard SFT continues to rely more heavily on larger dataset sizes for competitive in-domain performance.
> - For out-domain performance, DLoFT consistently outperforms SFT, even at smaller or larger data sizes.
>
> ---
>
> **Table 1: Comparison on the Meditron-7B model fine-tuned on the Medical-o1 dataset with different sizes.**
>
> | Benchmark | Method | Medical-o1-2.5K | Medical-o1-5K | Medical-o1-7.5K | Medical-o1-10K | Medical-o1-20K |
> |:-|:-|:-:|:-:|:-:|:-:|:-:|
> | **MedQA (in-domain)** | baseline   | 47.9 | 47.9| 47.9| 47.9 | 47.9 |
> | | +SFT | 50.1| 55.2 | 56.5 | 57.5 | 57.7 |
> | | +DLoFT | 57.3 | 57.8 | 57.8 | 58.2  | 58.2 |
> | **LiveCodeBench (out-domain)** | baseline | 18.4 | 18.4 | 18.4 | 18.4 | 18.4 |
> | | +SFT | 15.5 |16.3 | 15.6 |16.0 | 15.2 |
> | | +DLoFT | 24.5  | 25.5 | 26.0 | 25.7 | 26.0 |
> | **AIME24 (out-domain)**| baseline | 3.3  | 3.3 | 3.3 | 3.3 | 3.3 |
> | | +SFT | 6.7 | 3.3 | 3.3  | 6.7 | 3.3 |
> | | +DLoFT | 10.0 | 10.0 | 6.7 | 10.0 |10.0 |
>
> ---
>
> **Table 2: Comparison on the Qwen2.5-Coder-7B-Instruct model fine-tuned on the OpenThoughts-Code dataset with different sizes.**
>
> | Benchmark | Method| OpenThoughts-Code-2.5K | OpenThoughts-Code-5K | OpenThoughts-Code-7.5K | OpenThoughts-Code-10K | OpenThoughts-Code-20K |
> |:-|:-|:-:|:-:|:-:|:-:|:-:|
> | **LiveCodeBench (in-domain)** | baseline | 33.5 | 33.5 | 33.5  | 33.5 | 33.5 |
> | | +SFT | 34.0 | 34.3 | 35.0 | 35.6 | 35.8 |
> | | +DLoFT | 35.2 | 35.5 | 35.7 | 36.0 | 36.0 |
> | **AIME24 (out-domain)** | baseline | 10.0 | 10.0 | 10.0 | 10.0 | 10.0|
> | | +SFT | 10.0 | 13.3 | 13.3 | 16.7 | 13.3 |
> | | +DLoFT | 20.0 | 26.7  | 23.3 | 26.7 | 26.7 |
> | **MedQA (out-domain)** | baseline | 41.0 | 41.0 | 41.0   | 41.0 | 41.0 |
> | | +SFT | 40.8 | 39.2 | 40.0 | 39.6 | 39.0 |
> | | +DLoFT | 42.6 | 46.6 | 46.5 | 46.8 | 46.6 |
>
> ---
>
> **Table 3: Comparison on the Qwen2.5-Math-7B-Instruct model fine-tuned on the OpenR1-Math dataset with different sizes.**
>
> | Benchmark | Method | OpenR1-Math-2.5K | OpenR1-Math-5K | OpenR1-Math-7.5K | OpenR1-Math-10K | OpenR1-Math-20K |
> |:-|:-|:-:|:-:|:--:|:-:|:-:|
> | **AIME24 (in-domain)** | baseline | 16.7  | 16.7    | 16.7  | 16.7 | 16.7 |
> | | +SFT       | 20.0  | 23.3  | 23.3 | 23.3 | 26.7    |
> | | +DLoFT     | 23.3  | 26.7   | 26.7 | 26.7            | 30.0  |
> | **LiveCodeBench (out-domain)** | baseline | 27.7         | 27.7 | 27.7  | 27.7  | 27.7 |
> | | +SFT       | 29.2            | 31.9           | 32.2             | 32.0 | 31.4 |
> | | +DLoFT     | 34.8            | 35.3           | 35.0             | 35.5  | 35.5 |
> | **MedQA (out-domain)**   | baseline   | 43.5            | 43.5           | 43.5 | 43.5  | 43.5 |
> | | +SFT  | 42.0            | 40.4           | 41.7             | 40.5    | 40.3  |
> | | +DLoFT | 50.6            | 57.0           | 57.5             | 57.6  | 57.2   |

---

> > ### Comment · Reviewer_Dhab · 2025-08-06
> >
> > thanks for your responses. i maintain my positive view of this work, with hopes the manuscript itself will also be improved as you detail here

---

> > > ### Author Response · Authors · 2025-08-06
> > > **Thanks for your response and recognition**
> > >
> > > Dear Reviewer Dhab:
> > >
> > > We greatly appreciate your recognition and are thankful for your contributions to the academic community.
> > >
> > > We are more than happy to address any questions you may have and will make revisions based on your suggestions.
> > >
> > > Best regards,
> > >
> > > Authors

---

### Official Review · Reviewer_EP1B · 2025-07-03

**Clarity:** 2
**Significance:** 2
**Originality:** 3
**Rating:** 4
**Confidence:** 3

**Summary:**

**Summary**

This paper addresses an important problem in training LLMs for long chain-of-thought (LongCoT) reasoning: models trained via supervised fine-tuning (SFT) tend to overfit to problem-specific knowledge rather than learning generalizable reasoning patterns. This paper proposes a technique to  decompose gradients into paradigm-relevant and content-relevant components, updating models using only the former and show that this results in better out of distribution generalization.

**Questions:**

please see inline in the weaknesses section

**Ethical Concerns:**

["NO or VERY MINOR ethics concerns only"]

**Final Justification:**

I increased the score by +2. The authors addressed an extensive list of my questions. The justification is detailed in my comments exchanged with the authors.

**Limitations:**

yes

**Quality:**

2

**Strengths And Weaknesses:**

**Strengths**
1. The paper tackles a timely problem as the community is working towards scaling reasoning models.

2. The gradient decomposition method is creative and leverages structural properties of LongCoT data in an interesting way.


**Weaknesses**

1. All main experiments are performed with small subsamples of 5K datapoints. This leads to three direct questions: a. Do the advantages of this approach materialize when larger datasets are used? b. Can this approach result in worse performance when more data is used, due to loss of content specific signal? c. How does the performance of models trained with DLoFT using 5K samples compare to models trained with models trained with full dataset?

2. Figure 1b shows that SFT fails to induce LongCoT in out of distribution queries. At the cutoff point, DLoFT uses LongCoT for almost 80% of the queries and yet achieves exactly the same accuracy. This seems to indicate that even though CoT was being used, it had no performance advantage or actual generalization of reasoning.

3. The results in Figure 2 seem to be incorrect. Looking at Figure 1a, it is clear that SFT on 7b model results in slight increase in OOD queries as well at the early stopping point. Figure 2 shows an 8.4% drop.

4. While the core algorithm seems somewhat reasonable, there is neither any theoretical nor empirical analysis to show how the technique works.

5. Equation 3 shows that dot product of reference and full gradients need to be computed in order to compute the content gradient. How does this work when for non square weight metrics?

6. The method requires computing two forward passes and gradients per batch (full response and solution only), effectively doubling training cost compared to standard SFT. This would also double the memory usage.

---

> ### Author Response · Authors · 2025-08-01
> **Rebuttal**
>
> Thank you for your detailed and constructive comments and contribution to the academic community!
> Your feedback is of great significance to us in further improving our article. We will address each of your questions, and we hope to receive a higher rating from you if you are satisfied with our responses.
>
> ---------------------
>
> ### **Q1: Clarification on scalability and effectiveness of DLoFT on larger training dataset**
>
> Thank you for raising these questions. We provide the following clarifications and experimental results to address your concerns in detail:
>
> ---
>
> **(1) The reason why we used 5K samples in main experiments.**
>
> In our main experiments, we used a 5K-sample subset randomly drawn from the full LongCoT dataset. This choice was made after extensive preliminary experiments across different dataset sizes. We found that 5K represents an optimal trade-off between performance and training cost, where DLoFT achieves most of its performance benefits with minimal computational cost.
>
> To justify this design, we now include these experimental results (Tables 1–3 below in next block) that systematically compare model performance at different dataset sizes: 2.5K, 5K, 7.5K, 10K, and 20K. These experiments are conducted and evaluated across three domains: medical, coding, and math reasoning, using three different excellent instruction-tuned domain-specific models, respectively.
>
> These results clearly show that:
> - DLoFT demonstrates strong data efficiency and has a low data volume requirement. The performance improves noticeably when increasing from 2.5K to 5K, but plateaus or only slightly improves beyond 5K. This indicates that 5K is a data level at which performance and efficiency achieve a trade-off.
> - In contrast, standard SFT continues to rely more heavily on larger dataset sizes for competitive in-domain performance.
> - For out-domain performance, DLoFT consistently outperforms SFT, even at smaller or larger data sizes.
>
> ----
>
> **(2) Response to your question "a": Do the advantages of this approach materialize when larger datasets are used?**
>
> Yes.
>
> - ***out-domain comparison***: Our experiments with 7.5K, 10K, and 20K datasets confirm that DLoFT continues to yield significant gains in out-domain performance as data increases, often with even more pronounced improvements.
>
>     For example, in Table 1 below (in next block), on LiveCodeBench (out-domain evaluation):
>     - with 5K training data: DLoFT outperforms SFT by +9.2 (25.5 vs. 16.3).
>     - with fully 20K training data: DLoFT outperforms SFT by +10.8 (26.0 vs. 15.2).
>
> - ***in-domain comparison***: The in-domain performance gap between DLoFT and SFT tends to narrow slightly with larger datasets, as expected.
>
>      For example, in Table 1 below (in next block), on MedQA (in-domain evaluation):
>     - with 5K training data: DLoFT outperforms SFT by +2.6 (57.8 vs. 55.2)
>     - with fully 20K training data: DLoFT outperforms SFT by +0.5 (58.2 vs. 57.7)
>
>     This trend is reasonable: as dataset size increases, SFT receives broader problem coverage, improving its memorization-based in-domain performance. However, DLoFT remains advantageous due to its generalization benefits, driven by targeted learning of reasoning paradigms.
>
>     From a theoretical perspective, this aligns with our motivation: SFT tends to overfit to problem-specific knowledge, and its performance is sensitive to the coverage of training examples. In contrast, DLoFT decouples problem-specific signals, and focuses on learning topic-agnostic reasoning behaviors, making it more robust to domain shifts and less dependent on the breadth of training data.
>
> ---
>
> **(3) Response to your question "b": Can this approach result in worse performance when more data is used, due to loss of content specific signal?**
>
> We do not observe any performance degradation for DLoFT with larger datasets. Across all benchmarks and data sizes (see Tables 1–3 below in next block), DLoFT maintains or slightly improves its performance as data increases.
>
> This behavior is well-grounded in the design of DLoFT: since problem-specific gradients are filtered out during training, DLoFT’s performance gains arise purely from learning and applying general reasoning paradigms, particularly the LongCoT behaviors. These general reasoning patterns are not harmed by additional data, and in fact, more diverse examples of reasoning can help the model better internalize how LongCoT thinking applies across problems.
>
> In addition, it is precisely because DLoFT filters out the gradient component related to problem-specific knowledge, that its performance is less sensitive to data size. This property supports our goal: to efficiently learn general and generalizable reasoning patterns from a small amount of representative LongCoT data.
>
> In summary, DLoFT's performance is stable or improved at larger training data sizes, not because of problem-specific memorization, but due to stronger generalization in learning reasoning paradigms.

---

> ### Author Response · Authors · 2025-08-01
> **cont.**
>
> **(4) Response to your question "c": How does the performance of models trained with DLoFT using 5K samples compare to models trained with models trained with full dataset?**
>
> As Tables 1 and 2 below, we compare the performance of models trained on the full datasets (OpenThoughts-Code-20K and Medical-o1-20K) and those trained on the 5K random subsets. It can be found that:
> - DLoFT with 5K samples already achieves most of the gains achievable with the full dataset.
> - DLoFT with fully 20K samples achieves a slight improvement in both in-domain and out-domain evaluation. This highlights both the efficiency and scalability of our approach.
>
> For example,
> - in Table 1 below,
>     - on MedQA (in-domain evaluation): 57.8 (5K) vs. 58.2 (20K)
>     - on LiveCodeBench (out-domain evaluation): 25.5 (5K) vs. 26.0 (20K)
>     - on AIME24 (out-domain evaluation): 10.0 (5K) vs. 10.0 (20K)
> - in Table 2 below,
>     - on LiveCodeBench (in-domain evaluation): 35.5 (5K) vs. 36.0 (20K)
>     - on MedQA (out-domain evaluation): 46.6 (5K) vs. 46.6 (20K)
>     - on AIME24 (out-domain evaluation): 26.7 (5K) vs. 26.7 (20K)
>
> In conclusion, DLoFT exhibits strong generalization benefits even at small data scales, scales well with larger datasets, and avoids the pitfalls of overfitting to problem-specific knowledge. These results reinforce the practicality and robustness of our method in realistic settings with limited supervision.
>
> ---
>
> **Table 1: Comparison on the Meditron-7B model fine-tuned on the Medical-o1 dataset with different sizes.**
> |Benchmark|Method|Medical-o1-2.5K|Medical-o1-5K|Medical-o1-7.5K|Medical-o1-10K|Medical-o1-20K|
> |:----------------------|:----------|:---------------:|:-------------:|:---------------:|:--------------:|:--------------:|
> |**MedQA(in-domain)**|baseline|47.9|47.9|47.9|47.9|47.9|
> ||+SFT|50.1|55.2|56.5|57.5|57.7|
> ||+DLoFT|57.3|57.8|57.8|58.2|58.2|
> |**LiveCodeBench(out-domain)**|baseline|18.4|18.4|18.4|18.4|18.4|
> ||+SFT|15.5|16.3|15.6|16.0|15.2|
> ||+DLoFT|24.5|25.5|26.0|25.7|26.0|
> |**AIME24(out-domain)**|baseline|3.3|3.3|3.3|3.3|3.3|
> ||+SFT|6.7|3.3|3.3|6.7|3.3|
> ||+DLoFT|10.0|10.0|6.7|10.0|10.0|
>
> ---
>
> **Table 2: Comparison on the Qwen2.5-Coder-7B-Instruct model fine-tuned on the OpenThoughts-Code dataset with different sizes.**
> |Benchmark|Method|OpenThoughts-Code-2.5K|OpenThoughts-Code-5K|OpenThoughts-Code-7.5K|OpenThoughts-Code-10K|OpenThoughts-Code-20K|
> |:----------------------|:----------|:---------------:|:-------------:|:---------------:|:--------------:|:--------------:|
> |**LiveCodeBench(in-domain)**|baseline|33.5|33.5|33.5|33.5|33.5|
> ||+SFT|34.0|34.3|35.0|35.6|35.8|
> ||+DLoFT|35.2|35.5|35.7|36.0|36.0|
> |**AIME24(out-domain)**|baseline|10.0|10.0|10.0|10.0|10.0|
> ||+SFT|10.0|13.3|13.3|16.7|13.3|
> ||+DLoFT|20.0|26.7|23.3|26.7|26.7|
> |**MedQA(out-domain)**|baseline|41.0|41.0|41.0|41.0|41.0|
> ||+SFT|40.8|39.2|40.0|39.6|39.0|
> ||+DLoFT|42.6|46.6|46.5|46.8|46.6|
>
> ---
>
> **Table 3: Comparison on the Qwen2.5-Math-7B-Instruct model fine-tuned on the OpenR1-Math dataset with different sizes.**
> |Benchmark|Method|OpenR1-Math-2.5K|OpenR1-Math-5K|OpenR1-Math-7.5K|OpenR1-Math-10K|OpenR1-Math-20K|
> |:----------------------|:----------|:---------------:|:-------------:|:---------------:|:--------------:|:--------------:|
> |**AIME24(in-domain)**|baseline|16.7|16.7|16.7|16.7|16.7|
> ||+SFT|20.0|23.3|23.3|23.3|26.7|
> ||+DLoFT|23.3|26.7|26.7|26.7|30.0|
> |**LiveCodeBench(out-domain)**|baseline|27.7|27.7|27.7|27.7|27.7|
> ||+SFT|29.2|31.9|32.2|32.0|31.4|
> ||+DLoFT|34.8|35.3|35.0|35.5|35.5|
> |**MedQA(out-domain)**|baseline|43.5|43.5|43.5|43.5|43.5|
> ||+SFT|42.0|40.4|41.7|40.5|40.3|
> ||+DLoFT|50.6|57.0|57.5|57.6|57.2|

---

> ### Author Response · Authors · 2025-08-01
> **cont.**
>
> ### **Q2: Clarification on the relationship between "emergence of LongCoT behavior" and "problem-solving accuracy" (in Figure 1b)**
>
> We would like to clarify that:
> - Figure 1b is used to demonstrate that, compared with SFT, our method DLoFT is faster and more effective at inducing LongCoT behavior, especially on out-domain queries. This behavioral shift is an essential first step toward learning the LongCoT reasoning paradigm.
>
> - However, it's important to note that the presence of LongCoT behaviors do not guarantee the correctness of the final answer. LongCoT behaviors refer to the model engaging in reflective reasoning, revisiting earlier steps, and exploring alternative solution paths. These behaviors are beneficial for problem-solving but do not deterministically lead to correct answers. A correct final answer is only achieved when all the reflective steps successfully identify and correct the errors, and the exploratory steps finally find a correct solution path. Therefore, the emergence of LongCoT behaviors increases the likelihood of correct reasoning but is not a sufficient condition for correctness.
>
> In summary, DLoFT’s ability to induce LongCoT reasoning behaviors earlier and more frequently is a key strength.
> ***The emergence of LongCoT behaviors is only the initial success of the learning process***.
> ***The model needs to further learn how to use LongCoT behaviors more effectively to achieve correct answer***.
> ***Although the accuracy may initially not increase, it sets the stage for further gains as the model continues to optimize how these behaviors are utilized for complex reasoning***.
> As shown in Figure 1b, even after 100% of test queries exhibit LongCoT behavior, the test accuracy continues to improve as training progresses. This indicates that inducing LongCoT behavior is a necessary step, but not the end goal — the model continues to refine how it applies this reasoning paradigm to solve the problem.
>
> ---
> ### **Q3: Clarification on the difference of results in Figure 1 and Figure 2**
>
> We confirm that the results in Figure 2 are correct. The inconsistent accuracy in MedQA between Figure 1(left) and Figure 2(upper) is attributed to differences in the training datasets used in the two experiments, therefore the SFT results naturally diverge across these two figures. In detail,
> - In Figure 1, we finetune Qwen2.5-7B-Instruct on OpenR1-Math-5K dataset (only math-domain)
> - In Figure 2(upper), we finetune Qwen2.5-7B-Instruct on s1K dataset (multiple domains)
>
> Note: The SFT performance reported in Figure 2 does not correspond to the early stopping point. In practice, it is difficult to find a consistent early stopping point for all out-domain benchmarks, because different out-domain benchmarks start to degrade at different times, and early stopping leads to insufficient learning in the in-domain.
>
> To make clear comparison, we provide the comparison on all evaluation benchmarks with Figure 1's setting as Table 5 below.
> For convenience, we also list the results in Figure 2(upper) in Table 4 below.
> - From Table 5 below, SFT trained on a single-domain dataset (math) suffers from even more severe generalization drops in many out-domains. In contrast, DLoFT consistently achieves strong out-domain performance regardless of training domain, highlighting its robustness and ability to capture domain-agnostic reasoning patterns.
> - These results support our core claim: DLoFT learns reasoning paradigms rather than memorizing problem-specific knowledge, thus being less sensitive to domain coverage in the training data and focusing on learning problem-agnostic reasoning patterns.
>
> ---
> **Table 4: Comparison on the Qwen2.5-7B-Instruct model fine-tuned on the s1K dataset. (same as Figure 2(upper))**
>
> ||math|physics|chemistry|biology|code|medicine|computer|electronic|communication|mechanical|astronomy|geography|civil|agriculture|economics|history|law|literature|philosophy|sociology|
> |:-:|:-:|:-:|:-:|:-:|:-:|:-:|:-:|:-:|:-:|:-:|:-:|:-:|:-:|:-:|:-:|:-:|:-:|:-:|:-:|:-:|
> |SFT (compared to baseline)|+13.3|+4.0|+6.0|+2.0|-1.9|-8.4|+4.0|+0.0|-16.0|-10.0|+6.0|-6.0|-4.0|-6.0|-2.0|-10.0|-8.0|0|+4.0|-4.0|
> |DLoFT (compared to baseline)|+16.7|+20.0|+10.0|+14.0|+2.0|+7.2|+14.0|+12.0|+2.0|+10.0|+10.0|+6.0|+6.0|+2.0|+10.0|+8.0|+12.0|+12.0|+14.0|+8.0|
>
> ---
> **Table 5: Comparison on the Qwen2.5-7B-Instruct model fine-tuned on the OpenR1-Math-5K dataset. (the same setting as Figure 1)**
>
> ||math|physics|chemistry|biology|code|medicine|computer|electronic|communication|mechanical|astronomy|geography|civil|agriculture|economics|history|law|literature|philosophy|sociology|
> |:-:|:-:|:-:|:-:|:-:|:-:|:-:|:-:|:-:|:-:|:-:|:-:|:-:|:-:|:-:|:-:|:-:|:-:|:-:|:-:|:-:|
> |SFT (compared to baseline)|+16.7|+0.8|0.0|-1.8|+1.2|-8.6|+6.0|+0.5|-12.0|+3.5|+3.0|+2.0|-4.0|-6.5|+5.5|-12.0|-2.0|-2.5|+3.0|-6.5|
> |DLoFT (compared to baseline)|+16.7|+18.5|+10.5|+15.0|+2.2|+6.3|+13.5|+12.0|+1.5|+8.0|+9.0|+7.0|+5.8|+2.0|+10.4|+8.0|+12.0|+11.5|+13.6|+8.0|

---

> ### Author Response · Authors · 2025-08-01
> **cont.**
>
> ### **Q4: Response to concern about theoretical or empirical analysis**
>
> We would like to clarify the core motivation, design rationale, and empirical evidence supporting our method.
>
> ***(1) Goal & Motivation: Learning LongCoT reasoning efficiently***
>
> - Our overarching goal is to enable large language models (LLMs) to acquire Long Chain-of-Thought (LongCoT) reasoning — an advanced reasoning paradigm that involves iterative reflection, exploration of alternative solutions, and deep deliberation.
> - LongCoT embodies a general-purpose reasoning pattern that is problem-agnostic and potentially transferable across diverse domains and problems. Given its generality, we believe that acquiring the LongCoT reasoning paradigm should not require training the model from scratch (all post-training procedures again), nor massive large-scale datasets (million-level). Instead, our goal is to teach a CoT-style instruction-tuned model (e.g., Qwen2.5-Instruct or LLaMA-Instruct series) the LongCoT reasoning pattern by fine-tuning it on a small set of representative LongCoT examples.
> - However, in practice, we observed that standard SFT fails to achieve this goal. As analyzed in Introduction section (lines 38–47), Figure 1, and Supplementary Material Section 3, SFT does learn behaviors associated with LongCoT reasoning (e.g., Figure 1b shows that SFT-trained models can exhibit LongCoT behavior even on out-of-distribution queries), but it also suffers from a consistent degradation in out-of-domain performance. Our analysis identifies the reason as overfitting to problem-specific information in the limited training set — leading to memorization that harms generalization.
> - This contradicts the motivation behind our setup: if LongCoT reasoning is indeed general-purpose and transferable, then learning it should not require large-scale data or exhaustive data coverage.
> - To address this, we propose DLoFT, a method that explicitly disentangles the learning of general reasoning patterns from memorization of problem-specific content. Inspired by the observation that SFT tends to entangle both aspects into a single gradient update, we design a gradient decoupling mechanism to extract the paradigm-relevant component of the gradient (associated with general LongCoT reasoning pattern) and discard the problem-specific component. This enables the model to acquire general and transferable reasoning strategies without being biased toward the problem-specific information.
>
> ***(2) Design Principle: Leveraging the LongCoT format to achieve paradigm-specific gradient decoupling***
>
> Our core insight is that the structural format of LongCoT data itself provides a natural signal for isolating the reasoning-paradigm-specific gradient.
> Specifically, each LongCoT response is composed of two distinct parts (as demonstrated in Section 3.1). Crucially, the thinking part encodes both the problem-specific information and LongCoT reasoning paradigm, and the solution part only reflects the problem-specific information.
>
> This natural separation gives us a unique opportunity: We can define two supervision targets from the same training sample: one is the full LongCoT response (thinking + solution) and the other is a reference response (solution part only). By computing the gradients with respect to both targets respectively, we can project out the component of the gradient that is aligned with problem-specific information, thereby isolating the residual gradient signal that captures what is unique to the LongCoT reasoning pattern (see details in Section 3.2 and 3.3).
>
> This gradient decoupling is not heuristic—it’s explicitly grounded in the format of the training data and structure of the learning signal.
>
> ***(3) Empirical Evidence: Demonstrating the Effectiveness of Gradient Decoupling for General Reasoning***
>
> We provide empirical analysis across multiple sections of our paper that supports the claim that our gradient decoupling strategy successfully captures and transfers problem-agnostic reasoning patterns.
>
> Our key evaluation criteria focus on both in-domain and out-domain benchmarks: if the decoupled gradient truly captures the LongCoT-style reasoning behaviors independent of specific question types or topics, then models trained with these gradients should exhibit strong reasoning ability even on out-domain topics beyond the LongCoT training dataset.
>
> Extensive experiments demonstrate that:
> - Our DLoFT effectively mitigates the overfitting problem of SFT and learns stronger generalizable LongCoT reasoning ability. For example, it significantly outperforms SFT by an average of +16.4 points on out-domains and has a notable advantage of +11.5 points on in-domains (see Figure 2).
> - As a cold-start stage for Reinforcement Learning, our DLoFT can free the cold-start from the dependence on in-domain LongCoT data, and serves as a stronger foundation for subsequent RL, amplifying the outcome of RL (see Figure 3).

---

> ### Author Response · Authors · 2025-08-01
> **cont.**
>
> ### **Q5: Response to concern about how do dot products work with non-square weight matrices**
>
> This question highlights a potential point of confusion in our notation, and we would like to clarify it.
>
> - Our implementation indeed involves computing the dot product between the full gradient $g_{\text{full}}$ and the reference gradient $g_{\text{ref}}$ (as in Eq.(3)).
> We clarify that all gradient tensors are flattened into vectors before performing any operations such as dot product or projection. This ensures that the gradient vectors are compatible in dimensionality, even when originating from weight matrices of arbitrary (non-square) shapes.
>
> - Formally, given any parameter tensor (e.g., a weight matrix of shape $d_{\text{out}} \times d_{\text{in}}$), its gradient is reshaped into a one-dimensional vector using `.view(-1)` (or `.reshape(-1)`) before participating in Eq.(3). The same transformation is applied to both $g_{\text{full}}$ and $g_{\text{ref}}$, so that their dot product $\langle g_{\text{full}}, g_{\text{ref}} \rangle$ is well-defined as an inner product between two vectors of the same length.
>
> We will make this clarification more explicit in the final version.
>
> ----
>
> ### **Q6: Clarification on the comparison of "training speed" and "memory usage" between SFT and DLoFT**
>
> Efficiency is an important factor.
> We have already discussed the efficiency comparison between SFT and DLoFT in Supplementary Material Section 4.4 from both qualitatively and quantitatively. For convenience, we place the relevant results of Table 4 in Supplementary Material in the table below.
>
> | Method | Running Time (per step) | GPU Memory (Average) | GPU Memory (Peak) |
> |:-:     |:-:                     |:-:                   |:-:                |
> | SFT    | 6.2 min                | 69986 MB               | 71312 MB            |
> | DLoFT  | 6.8 min                | 70684 MB               | 78504 MB            |
>
> - First, from the Table above, we can found that: DLoFT increases the per-step training time by only 0.6 minutes, and the average GPU memory usage rises by 698 MB, with the peak usage increasing by 7,192 MB. All these values are well within the capacity of commonly used GPUs (e.g., NVIDIA A100 80GB).
> - Second, we would like to explain why DLoFT does not double the cost, even though it introduces an additional forward and backward pass.
>     - While standard SFT involves a single forward and backward pass on the full response, DLoFT requires an additional forward and backward pass on the solution-only portion. However, in LongCoT responses, the solution part is typically much shorter than the reasoning (thinking) part. We analyzed the datasets used in this paper and found that the ratio of the token number in the solution part to that in the full response ranges from 0.009 to 0.37, with a mean of 0.09. Therefore, the solution part accounts for only a small fraction of the full LongCoT response.
>     - As we know, the computational complexity of self-attention increases quadratically with the length of the input sequence. Therefore, performing an additional forward and backward pass on the solution consumes far less time and GPU memory than performing a single forward and backward pass on the full response. Thus, the actual overhead is far less than 2× in both time and memory.
>
> In summary, although DLoFT includes an additional pass, the overhead in practice is minimal and remains practical for modern hardware. We hope this clarifies the concern.

---

> ### Author Response · Authors · 2025-08-07
>
> Dear Reviewer EP1B:
>
> Thank you very much for your time and effort during the initial review period.
>
> We have done our best to address your comments and questions in our rebuttal. As the discussion phase is nearing its end, we would greatly appreciate it if you could let us know whether our responses have sufficiently addressed your concerns. Of course, we are also more than happy to further clarify or discuss any remaining issues you may have.
>
> We look forward to hearing from you, and thank you again.
>
> Best regards,
>
> Paper 5911 Authors

---

> > ### Comment · Reviewer_EP1B · 2025-08-08
> > **working through this -- could you please help prioritize the key author comments?**
> >
> > Hi, thanks for such a massive response. For the record, a quick google doc count shows your aggregate response to be more than 20,000 characters, twice the limit allowed. So it's a lot to process (this is a 4 page pdf in small font).  I just wanted to let you know that I am working through it and will reply today. It will greatly help if you could help prioritize the most important parts of your response that you would fit in the 10,000 character limit if you had to. Thanks!
> >
> > Ref: https://neurips.cc/Conferences/2025/PaperInformation/NeurIPS-FAQ
> >
> > Q: What is the length limit in the rebuttal phase?
> >
> > A: We have stopped supporting the global rebuttal option and increased the per-review rebuttal limit to 10,000 characters accordingly. You may use plain text with markdown formatting supported by OpenReview, but you cannot upload any additional files.

---

> > ### Comment · Reviewer_EP1B · 2025-08-09
> > **rebuttal response**
> >
> > It took me quite some time to work through your response. Sorry for the delay (due to the large volume of rebuttal material).
> > Let me start with the part you care about the most --> I will +1 the score. You've answered most of the questions. At a high level:
> >
> > q1: reasonably well answered and supported with additional data.
> >
> > q2: reasonable explanation , thanks.
> >
> > q3: this is the main unanswered part (see below).
> >
> > q4: I think the authors conflate analysis with evidence in their answer.
> >
> > q5: my question was answered.
> >
> > q6: question was well addressed.
> >
> > Q3: Fig1 shows that at early stopping point, the perf difference between SFT and DLOFT is small. the gap emerges due to overfitting on domain samples. With that said, it doesn't mean that there is nothing interesting happening here at all. Your method does have ~2% higher total perf. And it's probably (likely) because CoT is helping in _some_ cases. However, because you are not using any early stopping and are training for a fairly large number of epochs (10 vs typically people use <5 epochs, though it depends on tons of hyper params etc), there's a possibility of overfitting. So we just have to take the numbers from Fig2 for granted and assume that the gains are actually due to your method and not due to overfitting. So Q3 remains my primary concern at this point.

---

> > > ### Author Response · Authors · 2025-08-09
> > > **Response for new comments**
> > >
> > > Dear Reviewer EP1B,
> > >
> > > We are truly grateful for the time and effort you have put into carefully reading through our previous responses — especially given the large volume of rebuttal material. We are also very glad that most of your concerns have been addressed, with only Q3 remaining open. And we sincerely appreciate your willingness to raise the score by +1.
> > >
> > > - Regarding Q3, we understand your concern and are currently making some preparations to provide a clearer response for you. This involves running and organizing further experimental results, which will take approximately one hour. Please wait for a moment, and thank you for your patience.
> > >
> > > - Here is a brief preview of what we will share:
> > >     - ***About epoch_number:*** Actually, we have already run experiments with "total_epoch=5" under the Figure 2 setting.
> > >         - On in-domain benchmarks (math, physics, chemistry, biology), "total_epoch=10" yields better results than "total_epoch=5" (especially for 7B-level model).
> > >         - However, on out-domain benchmarks, performance drops are already visible at "total_epoch=5" setting.
> > >         - This was the reason we adopted "total_epoch=10" as our final setting.
> > >     - ***About comparison under early stopping:*** From our observations in the "total_epoch=10" experiments, the performance drop points for different out-domain benchmarks vary greatly — in some benchmarks (especially for some fields far from in-domains, such as history and agriculture), the drop happens as early as the first epoch. This makes it extremely difficult to choose a single consistent early stopping point across all benchmarks. For this reason, we only shows the early stopping point for a single out-domain benchmark in Figure 1.
> > >     - ***Upcoming results:*** We will later share the comparison between SFT and DLoFT for Figure 2 (upper) with "total_epoch=5" setting, including the per-epoch checkpoint performance on each benchmark. This will allow you to directly see both the inconsistency of early stopping points across benchmarks and the performance comparison under "total_epoch=5".
> > >
> > >
> > > Best regards,
> > >
> > > Paper 5911 Authors

---

> ### Author Response · Authors · 2025-08-09
> **Further Response for Q3**
>
> Dear Reviewer EP1B,
>
> We sincerely thank you for your patience in waiting for our follow-up.
>
> We have provided additional experiments under "total_epoch = 5" and reported the results in the Table below for both SFT and DLoFT. We also report the per-epoch performance during training to examine the performance evolution.
>
> **Key findings (when total_epoch=5):**
> - ***Out-domain performance degradation still persists for SFT:***
> The average drop over the 16 out-domain benchmarks is -3.8 under total_epoch = 5, compared to -3.9 under total_epoch = 10. This shows that shortening training from 10 to 5 epochs only marginally reduces the degradation.
>
> - ***DLoFT still maintains a clear advantage over SFT:***
> Under total_epoch = 5, DLoFT still significantly outperforms SFT on both in-domain and out-domain benchmarks, indicating that our gains are not simply due to prolonged training.
>
> - ***Early stopping based on averaged out-domain performance is suboptimal:***
>     - If we use the average performance across the 16 out-domain benchmarks to determine an early stop point for SFT, the optimal stopping point would be at the end of epoch 3. However, at this point, the average out-domain performance is still lower than before training, even though some individual benchmarks improve. This suggests that a single early stopping point cannot fully prevent out-domain degradation for SFT.
>     - Our per-epoch checkpoint results (columns 1-5) confirm our earlier statement: the epoch at which out-domain performance starts to drop varies greatly across benchmarks — some see declines as early as epoch 1 (e.g., history, agriculture), while others remain stable for longer. This large variability makes it infeasible to choose a single consistent early stopping point that benefits all out-domain benchmarks simultaneously.
>
> We hope this new evidence can address your concern regarding Q3. If you still have any concerns, please feel free to contact us at any time — we would be more than happy to clarify further and will remain responsive until the end of the discussion phase.
>
> If our additional clarifications and experiments have resolved your concerns, we would be sincerely grateful if you could consider further increasing your final score judgment, moving it out of the reject score range. Thanks again.
>
> ---
> **Table: Performance Comparison on Qwen2.5-7B-Instruct model fine-tuned on the s1K dataset.**
>
> - Performance Format: SFT / DLoFT
> - We report the performance improvement relative to the model before training
> - Columns 2-5: Performance of the intermediate model checkpoints at 1st ~ 4th epoch during training
> - Column 6: Performance of the final checkpoints at the last epoch
> - Column 7: Performance of the final checkpoint at the last epoch
>
> | current_epoch (total_epoch)      | epoch_1 (5) | epoch_2 (5) | epoch_3 (5) | epoch_4 (5) | epoch_5 (5) | epoch_10 (10) |
> |-|-|-|-|-|-|-|
> | math | -3.3/+0.0  | +0.0/+6.7  | +6.7/+10.0 | +6.7/+16.7 | +10.0/+16.7 | +13.3/+16.7 |
> | physics | -0.0/+4.6  | +0.3/+7.4  | +2.7/+15.2 | +3.0/+16.8 | +3.4/+18.8 | +4.0/+20.0 |
> | chemistry | +0.6/+2.2  | +1.0/+1.0  | +1.1/+3.5  | +2.9/+8.2  | +3.6/+8.9  | +6.0/+10.0 |
> | biology | +0.4/+5.0  | +0.8/+7.8  | +1.0/+10.3 | +1.5/+12.0 | +1.8/+13.4 | +2.0/+14.0 |
> | code  | -0.3/+0.0  | +0.1/+0.4  | -1.6/+0.5  | -2.5/+1.7  | -2.4/+1.9  | -1.9/+2.0 |
> | medicine  | -0.4/+2.2  | -1.2/+3.6  | +2.3/+4.5  | -5.8/+5.9  | -6.2/+6.6  | -8.4/+7.2 |
> | computer | +0.0/+8.3  | +0.6/+11.0 | +0.3/+10.4 | +1.4/+13.3 | +1.1/+14.0 | +4.0/+14.0 |
> | electronic | -0.2/+5.4  | +0.1/+9.1  | -0.1/+8.8  | +0.4/+10.2 | +0.4/+12.3 | 0.0/+12.0 |
> | communication | -5.3/+0.1  | -3.6/+0.4  | -15.0/+1.4 | -13.3/+2.0 | -14.5/+2.2 | -16.0/+2.0 |
> | mechanical | -1.3/+4.3  | -4.5/+2.8  | -5.3/+6.2  | -11.7/+8.8 | -11.3/+9.4 | -10.0/+10.0 |
> | astronomy | +0.7/+0.5  | +0.0/+2.7  | +3.2/+6.3  | +4.6/+9.9  | +5.4/+10.5 | +6.0/+10.0 |
> | geography | -1.2/-0.2  | -2.0/+2.0  | +0.5/+4.5  | -4.3/+5.8  | -7.0/+6.0  | -6.0/+6.0 |
> | civil  | -3.3/+2.1  | -2.7/+4.2  | -1.4/+3.2  | -3.6/+5.3  | -4.3/+5.6  | -4.0/+6.0 |
> | agriculture | -8.2/+1.0  | -1.6/+2.4  | -5.3/+2.0  | -4.0/+1.8  | -6.6/+2.0  | -6.0/+2.0 |
> | economics | +0.2/+6.2  | -0.1/+5.9  | +0.0/+7.0  | +0.3/+9.5  | -0.2/+9.7  | -2.0/+10.0 |
> | history | -4.0/+3.3  | -7.8/+6.7  | -6.5/+4.0  | -6.1/+7.4  | -8.8/+7.5  | -10.0/+8.0 |
> | law | +0.5/+8.6  | -4.4/+10.0 | -5.2/+9.8  | -4.0/+11.2 | -3.5/+11.6 | -8.0/+12.0 |
> | literature  | 0.0/+2.8   | -3.3/+6.3  | +1.7/+8.8  | -1.0/+9.6  | -1.0/+11.5 | 0.0/+12.0 |
> | philosophy | +0.8/+4.0  | +1.1/+8.3  | +0.4/+6.6  | +0.8/+10.4 | +0.8/+13.9 | +4.0/+14.0 |
> | sociology | -1.3/+1.3  | -2.0/+2.0  | +0.2/+4.1  | -2.9/+6.7  | -3.3/+8.2  | -4.0/+8.0 |
> | avg (4 in-domain) | -0.6/+3.0 | +0.5/+5.7 | +2.9/+9.8 | +3.5/+13.4 | +4.7/+14.5 | +6.3/+15.2 |
> | avg (16 out-domain) | -3.8/+3.1 | -2.0/+4.9 | -1.3/+5.5 | -2.4/+7.5 | -3.8/+8.3 | -3.9/+8.5 |
> | avg (all) | -1.3/+3.1  | -1.9/+5.0  | -0.4/+6.4  | -1.2/+8.7  | -2.1/+9.5  | -1.8/+9.8 |

---

### Note · Authors · 2025-08-12

Dear ACs and Reviewers,

We sincerely thank you for your time and effort in evaluating our paper.
Below, we summarize the key points during rebuttal and discussion.

|Reviewer|Reviewer's Feedback for Rebuttal|Our Further Response|Reviewer's Further Feedback|
|-|-|-|-|
|Dhab|All concerns are addressed. Maintain positive view. Elegant approach. Strong Performance|not needed|not needed|
|bbCD|All concerns are addressed. No important weakness|not needed|not needed|
|EP1B|Remain one question: **(1)** Does performance gain of DLoFT come from SFT's overfitting? Request for comparison under less-likely overfitting setup: smaller epoch (e.g., 5) and early stopping|**(1)** Provide additional experiments to show significant gain of DLoFT than SFT (e.g., +12.1 when epoch=5, +9.6 compared to SFT with early stopping)|no reply (we believe it is resolved)|
|cC52|Remain four questions: **(1)** Doubt the training convergence, as the reported performance of SFT is less significant compared to reviewer's experience. **(2)** Request for theoretical proof to support "problem-specific knowledge hinders generalization". **(3)** DeepSeek-R1-Distill models demonstrate that SFT on LongCoT data can facilitate a broad range of reasoning tasks, which seems to contradict the failure of SFT claimed in this paper. **(4)** Small gains of SFT+RL are unreasonable since GRPO typically yield large gains|**(1)** Provide evidence to confirm full convergence as our reported result(49.9) aligns with official paper(50.0), with only 0.1 random variation. **(2)** Provide theoretical foundations (link to shortcut learning and spurious correlations in ML theory), similar conclusion from prior studies, and experimental results from our paper to strongly support "problem-specific knowledge hinders generalization" is not a baseless assumption. **(3)** Make explanation: The failure of SFT founded in our work is not in conflict with "the success of SFT in developing DeepSeek-R1-Distill". We target a different scenario—efficiently internalizing LongCoT ability into CoT instruction model without harming existing knowledge (where SFT fails). However, DeepSeek-R1-Distill trains from base model to simultaneously activate knowledge and learn LongCoT (where SFT works). **(4)** Highlight the substantial gain: For in-domain, SFT+RL achieves +13.3(math) and +11.0(medical) gains. For out-domain, RL still achieves +3.3(math) and +2.6(medical) gains, recovering part of SFT drop|no reply (we believe they are resolved)|

---

### Decision · Program_Chairs · 2025-09-17

**Decision:**

Accept (poster)

**Comment:**

This paper studies the out-of-distribution (OOD) generalization failures of naively fine-tuning LLMs on long Chain-of-Thought (CoT) data and proposes a simple yet effective method (DLoFT) to mitigate this issue. The core idea is to isolate general “long reasoning” abilities from "problem-specific" signals during SFT. This is achieved by projecting out the problem specific gradients (obtained from question -> solution prediction) from the total gradient (obtained from question -> (thoughts, solution) prediction). Overall this improves over vanilla SFT (even after some RL) across the board, particularly on OOD benchmarks.
Overall, this paper addresses the timely and important topic, identifies a crucial weakness in the standard SFT approach for LongCoT data and proposes a simple, effective, and well-motivated solution.

---

Reviewers raised a few key concerns, which the authors have largely addressed:
* **Overfitting and Dataset Size:** A primary concern was whether vanilla SFT was worse simply due to overfitting. The authors mitigated this by showing that training for fewer epochs and much larger SFT dataset shows similar trends.
* **SFT and RL numbers are low:** Authors pointed to the s1k paper and argued that RL showed reasonable improvements.
* **(Theoretical) justification for the decoupling** There were questions about why problem-specific features would hurt generalization and whether this contradicts recent work (e.g., DeepSeek-V2) suggesting SFT is beneficial. The authors pointed to literature on overfitting and shortcut learning for justification, and commented that since they are fine-tuning an instruction-tuned model, the results are not contradictory.

---

One issue (not discussed in the author-reviewer discussion) is that the paper lacks good baselines comparisons for the method. The paper mostly compares DLoFT to vanilla SFT, early stopping and weight decay, but perhaps some more baselines (e.g. in-context prompting with LongCoT data or LoRA fine-tuning) could have been considered for better generalization.
Furthermore, the problem being studied shares similarities with "catastrophic forgetting" in continual learning and also shortcut learning. In fact some gradient orthogonalization ideas exist in the continual learning literature [1]. At least a discussion with this literature is warranted, and perhaps some simple baselines from these areas would further contextualize the paper's contribution and strengthen the results.

---

Overall the paper makes a positive contribution, and incorporating the experiments and fixes suggested by the reviewers would further enhance the manuscript. The recommendation is Accept.

[1] Farajtabar et al. Orthogonal Gradient Descent for Continual Learning. 2019